# POWER AND LIMITATIONS OF AGGREGATION IN COMPOUND AI SYSTEMS

## ABSTRACT

When designing AI systems for complex tasks, it is becoming increasingly common to query a model in different ways and aggregate the outputs to create a compound AI system. In this work, we mathematically study the power and limitations of aggregation within a stylized principal-agent framework. This framework models how the system designer can partially steer each agent's output through reward specification, but still faces limitations due to prompt engineering ability and model capabilities. Our analysis identifies three mechanisms—feasibility expansion, support expansion, and binding set contraction—through which aggregation can expand the set of elicitable outputs. We prove that any aggregation operation must implement one of these mechanisms to provide benefit, though none are sufficient alone. To sharpen this picture, we establish necessary and sufficient conditions for when aggregation expands elicitable outputs. Altogether, our results take a step towards characterizing when compound AI systems can overcome limitations in model capabilities and in prompt engineering.

## 1 INTRODUCTION

Compound AI systems—which leverage multiple AI components, rather than a single model in isolation—present a powerful paradigm to tackle complex tasks (BAIR Research Blog, 2024). In the context of large language models (LLMs), one common approach is to create many copies of the same model, give these models different prompts or access to different tools, and aggregate the outputs of these models at test-time. This approach has proven fruitful in multi-agent research systems (Anthropic Engineering, 2024) where a lead LLM agent delegates subtasks to different specialized agents and aggregates their outputs, in multi-agent debate protocols where different LLM agents seek consensus (Du et al., 2024) or argue for different answers (Khan et al., 2024), and in prompt ensembling approaches where the outputs from different prompts are combined (Arora et al., 2023).

Given the empirical success of these compound LLM systems, this raises the question of when aggregating across multiple copies of the same model unlocks greater performance than querying a single model. At first glance, aggregation may seem redundant when the model copies are homogeneous. However, one source of improved performance is at the prompt level: a model with a complex prompt engineering approach may be replaceable by a set of models with simple but diverse prompting strategies (Arora et al., 2023), illustrating how aggregation across models can overcome limitations in prompt engineering ability. Another source of improved performance is at the output level: aggregating multiple LLM agents over repeated interactions can help correct errors such as hallucinations (Du et al., 2024), illustrating how aggregation can overcome limitations in model capabilities as well. This suggests that the extent to which aggregation overcomes these limitations in prompt engineering and model capabilities impacts the power of compound AI systems.

In this work, we study the power and limitations of aggregation from a theoretical perspective, building on a classical principal-agent framework (Kleinberg et al., 2019). Our focus is on compound AI systems where a system designer passes reward specifications (e.g., via prompts) to many copies of the same model and then aggregates their outputs. In this stylized principal-agent framework (Section 2), the system designer (i.e., the principal) designs reward specifications to elicit $N$-dimensional outputs from each agent, and aggregates these outputs to produce a synthesized output. Each agent generates the outputs in its feasible set that maximizes the reward, and the system-designer strategically co-designs the rewards across models to try to produce a specific output. We capture prompt

engineering limitations as the rewards operating over a coarser $M$-dimensional feature space, and model capability limitations as conic constraints on each agent's feasible set of outputs.

Using this framework, we characterize when aggregating across multiple agents enables the system designer to elicit to a greater set of outputs than relying on a single model. To build intuition, we formalize three natural mechanisms by which aggregation can expand the set of elicitable outputs (Section 3). The first mechanism is *feasibility expansion*, where aggregation produces outputs outside of any agent's feasibility set. The second is *support expansion*, where aggregation combines outputs with smaller supports into an output with a larger support. The third is *binding set contraction*, where aggregation combines outputs that are binding with respect to constraints into an output that falls within the interior.

We formally connect these mechanisms to elicitability-expansion. Specifically, we find that the power of aggregation fundamentally relies on at least one of these mechanisms being implemented: if none are implemented, then aggregation does not expand elicitability on any problem instance (Theorem 3.7). However, these mechanisms are not sufficient to expand elicitability in general, although we show that each mechanism results in elicitability-expansion under stronger conditions.

To more completely capture the power and limitations of aggregation, we provide a more general characterization of elicitability-expansion (Section 4). We first characterize when an aggregation operation is elicitability-expanding in a given problem instance (Theorem 4.1), linking this to whether feasible directions for agent outputs intersect with feature-improving directions. To analyze the limitations of aggregation, we derive general conditions (Definition 4.2) under which an aggregation operation never expands the set of elicitable outputs, regardless of the level of coarseness of the feature space (Theorem 4.3), and we show that these conditions are tight (Theorem 4.4). These tight conditions in Definition 4.2 are strengthenings of feasibility expansion, support expansion, and binding-set contraction. At a high-level, these conditions test whether feasible directions under which an agent can change the aggregated output violate the constraints by a sufficient margin.

Altogether, our results uncover key mechanisms that underpin the power and limitations of an aggregation in compound AI systems. Our results suggest conditions for aggregation to add no power to a system, regardless of the level of prompt engineering limitations. Moreover, our results illustrate how the power of an aggregation depends on the interplay between prompt engineering ability and model capabilities. More broadly, our results take a step towards understanding when aggregation of multiple copies of the same model provides benefits to system designers.

## 1.1 RELATED WORK

**Aggregation across multiple models.** Aggregating outputs from multiple LLMs is a common strategy for complex tasks (BAIR Research Blog, 2024). One common approach is resampling the same model or reasoning trace and then selecting outputs via reward models (Christiano et al., 2017), self-consistency (Wang et al., 2023b), or synthesis (Zhang et al., 2025); coverage is an important property for inference-time computations (Huang et al., 2025). Other approaches are routing queries across different LLMs (Chen et al., 2024), adversarially combining models to expose safety risks (Jones et al., 2025a), and consensus games between generators and discriminators (Jacob & Andreas, 2024). Closest to our setting are systems with multiple copies of the same model under different reward specifications, as in LLM debate (Du et al., 2024), prompt ensembling (Arora et al., 2023), and multi-agent research frameworks (Anthropic Engineering, 2024). We provide a theoretical perspective on when such aggregation elicits strictly more outputs than a single model. Classical work has analyzed aggregation in settings such as ensembling (Dietterich, 2000), voting (Ladha, 1992), distributed algorithms (Lynch, 1996), and multi-agent reinforcement learning (Tan, 1993).

**Principal-Agent Models and Reward Design.** Our model is inspired by the principal-agent model by Kleinberg et al. (2019). We extend their technical result to incorporate agent limitations in the form of conic constraints and derive new results that characterize elicitability via aggregation. This falls under the broader principal-agent framework (Holmström, 1979; Grossman & Hart, 1983; Laffont & Martimort, 2002; Bolton & Dewatripont, 2005), which captures the challenge of designing rewards based on imperfect proxies. (Zhuang & Hadfield-Menell, 2020) use this framework to study misalignment of AI, which is similar to our motivation. Work in this framework also incorporates agent's limitations in the form of costs for actions. Particularly related are multitask settings (Holmström & Milgrom, 1991; Slade, 1996; Bond & Gomes, 2009; Demougin et al., 2022) that study

the effects of costs being dependent between tasks, including cases of substitutability and complementarity, which is similar to our conic constraints that capture dependence among multiple output dimensions. Principal–agent theory has also considered multiple agents (Holmström, 1982; Lazear & Rosen, 1981; Dasaratha et al., 2024), focusing mainly on the joint design of rewards. Our work differs in allowing aggregation to synthesize new outputs, and in characterizing when aggregation provides provable benefits rather than addressing algorithmic design. Complementary work studies benefits of heterogeneity across agents (Gentzkow & Kamenica, 2017; Collina et al., 2025), though they don't study heterogeneity through differently designed rewards.

## 2 MODEL

We extend the principal-agent framework in Kleinberg et al. (2019) to model a compound AI system with $K$ agents (who represent LLMs) and a single principal (the system designer). The system designer designs reward specifications to elicit outputs from the agents, and aggregates the outputs to synthesize a new output. The system designer faces limitations on the complexity of rewards they can design, and the agents face limitations in terms of the space of outputs that they can generate. We defer a discussion of model limitations to section 5.

### 2.1 OUTPUT SPACE

We embed outputs of agents into $M$-dimensional vectors with non-negative coordinates. We view each output dimension as capturing a different characteristic of the output. The vector representation $\boldsymbol{x}$ quantifies the degree to which the output captures each characteristic. We note that some dimensions may capture undesirable characteristics (e.g., hallucinations). The system designer seeks a specific output $\boldsymbol{x}^{(A)} \in \mathbb{R}_{\geq 0}^M$, which we assume to be unit $\ell_1$-norm $\|\boldsymbol{x}^{(A)}\|_1 = 1$.

Our model captures how the agents have restrictions on the set of output vectors that it can produce, for example due to capability limitations. The first restriction is that the $\ell_1$ norm of the output vectors is bounded, which captures budget limitations. The second restriction is conic constraints on the output, which each take the form $\mathbf{c}^T \boldsymbol{x} \leq 0$ where $\mathbf{c} \in \mathbb{R}^M$ contains at least strictly positive entry and at least strictly negative entry. These conic constraints capture restrictions on the types of outputs that the agent can produce: for example, some agents may not be able to avoid producing hallucinations without facing capability degradation along other characteristics.

We let $L$ denote the number of conic constraints, and we let $\boldsymbol{C} \in \mathbb{R}^{L \times M}$ denote the conic constraints themselves. Let $\boldsymbol{C}_i \in \mathbb{R}^M$ denote the $i$th row of $\boldsymbol{C}$ for $i \in [L]$, and let $\boldsymbol{C}_V \in \mathbb{R}^{|V| \times M}$ denote the set of rows corresponding to indices $V \subseteq [M]$. We denote by $\boldsymbol{C}_\varnothing$ the zero-vector, to capture how $\{\boldsymbol{d} : C_\varnothing \leq 0\} = \mathbb{R}_{\geq 0}^M$. Given a budget level $E > 0$, we let $\mathcal{B}(E)$ denote the feasible set at budget level $E$, defined to be:

$$\mathcal{B}(E) := \left\{ \boldsymbol{x} \in \mathbb{R}_{\geq 0} \mid \boldsymbol{C}\boldsymbol{x} \geq \boldsymbol{0}, \|\boldsymbol{x}\|_1 \leq E \right\}.$$

### 2.2 REWARD SPECIFICATION

The system designer designs a reward specification $R^{(k)}$ and a budget level $E^{(k)}$ for each agent $k \in [K]$. The reward specification represents the reward implicit in the prompt that they give to the agent, and the budget level represents the level of test-time compute that the agent is allowed to use.

To capture prompt engineering limitations, we model the reward specification as operating over a coarser $N$-dimensional feature space than the outputs. Here, the features $F(\boldsymbol{x}) = [F_1(\boldsymbol{x}), \ldots, F_N(\boldsymbol{x})]$ take the form

$$F_j(\boldsymbol{x}) = f_j\left(\sum_{i=1}^M \alpha_{ij}\boldsymbol{x}_i\right),$$

where $f_j(\cdot)$ is nonnegative, smooth, weakly concave (i.e., diminishing returns from increasing quality on this dimension), and strictly increasing, and where the values $\alpha_{ij} \geq 0$ are nonnegative *feature weights*. We will denote by $\boldsymbol{\alpha} \in \mathbb{R}_{>0}^{M \times N}$ the matrix with entries $\alpha_{ij}$ and call this the *feature weights matrix*.

We consider reward specifications $R^{(1)}, \ldots, R^{(K)} : \mathbb{R}^N \to \mathbb{R}$ which operate on these features. Following prior work (Kleinberg et al., 2019), we restrict to *monotone* reward functions $R$ which do not decrease if all features are weakly increased, and where there exists $j \in [N]$ such that $R$ strictly increases whenever the feature $F_j$ strictly increases.

Given a monotone reward specification $R^{(k)}$ and a positive budget level $E^{(k)} > 0$, each agent $k$ produces an output that maximizes its reward over the feasible set $\mathcal{B}(E^{(k)})$: that is,

$$\boldsymbol{x} \in \mathbf{X}^*(R^{(k)}, E^{(k)}) := \operatorname{argmax}_{\boldsymbol{x} \in \mathcal{B}(E^{(k)})} R^{(k)}(F(\boldsymbol{x})).$$

This captures how even though agents are homogeneous and solve the same optimization program, they can be given different reward specifications and thus produce different outputs.

### 2.3 Elicitability

We say that a reward specification $R$ and budget level $E$ elicits an output $\boldsymbol{x}$ if $\boldsymbol{x} \in \mathbf{X}^*(R, E)$. This captures whether an agent can produce the output $\boldsymbol{x}$: that is, if $\boldsymbol{x} \in \operatorname{argmax}_{\boldsymbol{x} \in \mathcal{B}(E)} R(F(\boldsymbol{x}))$. As shown in prior work (Kleinberg et al., 2019) and illustrated in Section 3.1, some output vectors $\boldsymbol{x} \in \mathcal{B}$ where $\boldsymbol{x}$ are not elicitable by any reward specification $R$ and budget level $E$.

We say that an output $\boldsymbol{x}$ is elicitable if there exists a monotone reward specification $R$ and a positive budget level $E$ that elicits $\boldsymbol{x}$. The condition for whether $\boldsymbol{x}$ is elicitable only depends on $\boldsymbol{x}$ through the following sufficient statistic $(\mathcal{S}(\boldsymbol{x}), \mathcal{V}(\boldsymbol{x}))$. The first component $\mathcal{S}(\boldsymbol{x}) = \{j : x_j > 0\}$ denotes the support of $x$. The second component $\mathcal{V}(\boldsymbol{x}) = \{l \in [L] : C_l \boldsymbol{x} = 0\}$ denotes the set of indices of conic constraints that are binding at $\boldsymbol{x}$.

**Aggregation.** When the system designer can aggregate the outputs of different agents, this may expand the set of elicitable outputs. The following definition captures when this occurs.

**Definition 2.1.** *We call $\boldsymbol{x}^{(1)} \ldots, \boldsymbol{x}^{(K)} \to \boldsymbol{x}^{(A)}$ is an **elicitability-expanding operation** if*

- *There exist monotone reward specifications $R^{(1)}, \ldots, R^{(K)}$ and positive budget levels $E^{(1)}, \ldots, E^{(K)}$ such that $\boldsymbol{x}^{(k)} \in X^*(R^{(k)}, E^{(k)})$ for all $k \in [K]$.*
- *There does not exist a monotone reward specification $R$ and budget level $E > 0$ such that $x^{(A)} \in X^*(R, E)$.*

Intuitively, if an aggregation operation is elicitability-expanding, then allowing the system-designer to aggregate outputs according to this operation produces an output that is not elicitable with a single reward, but can be obtained by combining outputs elicited from multiple reward specifications.

**Aggregation rules.** An aggregation rule is a mapping from a list of output vectors $(\boldsymbol{x}^{(1)}, \ldots, \boldsymbol{x}^{(K)})$ to an aggregated output vector $\boldsymbol{x}^{(A)}$. There are two natural aggregation rules we will often consider in our work. Although our results apply to more general aggregation rules, we will often use these natural aggregation rules to provide examples.

The first is *intersection aggregation*, which is defined to be the coordinate-wise minimum of the vectors:

$$\mathcal{A}_{\text{intersect}}(\boldsymbol{x}^{(1)}, \ldots, \boldsymbol{x}^{(K)}) = \boldsymbol{x}^{(1)} \wedge \ldots \wedge \boldsymbol{x}^{(K)}. \tag{1}$$

This aggregation rule combines outputs based on commonality among different output vectors, which is conceptually similar to debate protocols (Du et al., 2024) that aim to create agreement or inference scaling methods that aim to filter out incorrect information (Zhang et al., 2025). The second is *addition aggregation*, which takes a weighted sum of the vectors. For a weight vector $\boldsymbol{w} \in \mathbb{R}_{\geq 0}^K$, the rule is given by

$$\mathcal{A}_{\text{add}}(\boldsymbol{x}^{(1)}, \ldots, \boldsymbol{x}^{(K)}; \boldsymbol{w}) = \sum_{i=1}^K \boldsymbol{w}_i \boldsymbol{x}^{(i)}. \tag{2}$$

Addition aggregation interpolates among different output directions. This rule conceptually captures system designers synthesize multiple outputs to delegate specialized subtasks to each agent and synthesize the outputs of these subtasks (BAIR Research Blog, 2024; Anthropic Engineering, 2024).

### 2.4 Illustrative Example: Citations Task

To ground our framework in a concrete setting, we consider a natural aggregation task—generating a list of papers on a given topic (Wang et al., 2023a; Press et al., 2024). We describe the task and then show how instantiations of our framework capture different aggregation behaviors for it.

**Task and Setup.** We study a citation task where the system designer seeks a list of 10 influential LLM papers spanning five perspectives: (1) ML theory, (2) NLP/CL, (3) cognitive science, (4) AI alignment and human–AI interaction, and (5) multi-agent systems.

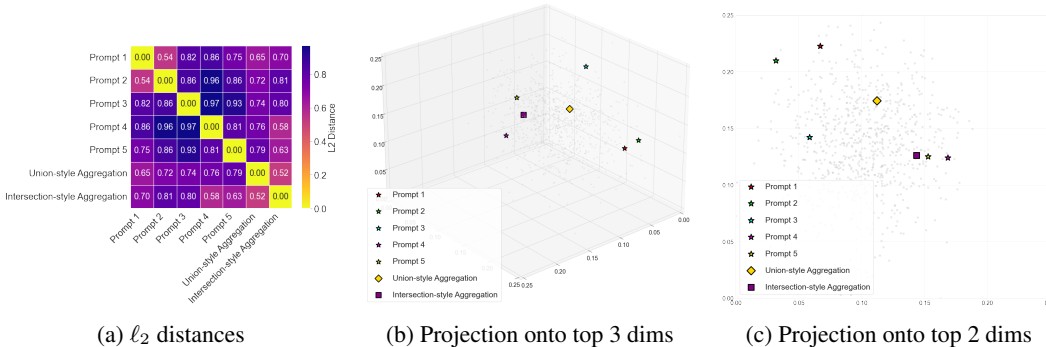

(a) $\ell_2$ distances      (b) Projection onto top 3 dims      (c) Projection onto top 2 dims

Figure 1: Visualization of output vectors for the citation generation task (Section 2.4). Output vectors are computed using the 768-dimensional embeddings from all-mpnet-base-v2, shifted to be in the nonnegative orthant. Embeddings are shown for GPT-4o-mini outputs from five different prompts, and as well as two different aggregated outputs based on additional-style and intersection-style aggregation rules. The $\ell_2$-distances (left) and projections onto the top 3 highest-variance (middle) and top 2 highest-variance dimensions (right), are shown. The plots show that the five prompts produce semantically different outputs, and each aggregation operation results in a combination of the five outputs that does not resemble any output in isolation.

The system designer issues five prompts, each targeting one perspective, to gpt-4o-mini-2024-07-18 and then aggregates the resulting lists. We prompt another LLM (also gpt-4o-mini-2024-07-18) to aggregate these five lists, instantiating aggregation rules that are inspired by intersection aggregation $\mathcal{A}_{\text{intersect}}$ and union aggregation $\mathcal{A}_{\text{add}}$. Specifically, the model is prompted with *aggregation instructions* along with the five different lists of 10 references, and produces an aggregated list of 10 references. The *intersection-style aggregation instructions* ask for references which are central and broadly relevant across all five perspectives, thus approximating intersection even when the literal overlap of references is empty. The *addition-style* aggregation instructions ask for references that jointly cover and reflect the combined topical space of all five perspectives. We defer the specific prompts and other details of the empirical setup to Appendix G.

**Output vectors.** We specify two different instantiation of output vectors in our framework, depending on the level of specificity of the output which the system designer aims to elicit.

1. Suppose that the system designer aims to elicit an output that balances multiple high-level criterion in a specific manner (e.g., covering papers in different subareas, covering up to the state of the art in each subarea, quality of the papers selected, etc.). To capture this, let each dimension of the output capture a different criterion that the system designer cares about. We can think of the value of the output vector along each dimension as the extent to which the output captures the criterion corresponding to that dimension.
2. Suppose that the system designer aims to elicit a specific output (e.g., a specific list of references). To capture this, we represent each model output as a high-dimensional embedding coming from a text embedding model. For the citation task, we use all-mpnet-base-v2 (Reimers & Gurevych, 2019), a sentence-transformers model that produces 768-dimensional vectors.[1] Figure 1 shows embeddings for outputs to the five prompts, as well as the outputs produced by the intersection-style and addition-style aggregation rules. The five prompt outputs vary substantially, and the aggregated outputs differ markedly from both the originals and from each other, demonstrating how different prompts and aggregation rules can reshape the embedding-space representation.

**Reward specification limitations.** Our framework captures two different types of reward specification limitations. First, the system designer may struggle to precisely express what they truly want in the prompt, leading to underspecified prompts omitting some of the system designer's requirements (Yang et al., 2025). For example, the system designer may prompt the model to include a "breadth

---

[1]As detailed in Appendix G, we apply an additive shift to ensure nonnegativity, computed from the minimum value in each dimension across 805 gpt-4o-mini-2024-07-18 outputs from the helpful-base AlpacaEval dataset (Li et al., 2023).

of citation coverage" when in reality they would like to restrict citations to a handful of academic venues. Second, the model may not correctly interpret the system designer's prompt by mapping two dissimilar words in the prompt to the same word (Jones et al., 2025b). In the citation task, this could surface as the model interpreting "papers with high attribute 'X'" similarly for many different attributes "X". We capture both of these forms of limitations as the reward specification operating over coarsenings of the output dimensions (as captured by the features) rather than directly on the output dimensions.

## 3 NATURAL MECHANISMS FOR ELICITABILITY-EXPANSION

In this section, we formalize natural mechanisms by which aggregation expands elicitability. First, we show how mechanisms expand elicitability via examples (Section 3.1). Then, we show that these mechanisms are necessary for elicitability-expansion (Section 3.2). The results in this section leverage the technical tools that we develop in Section 4. Note that our goal in this section is to link the mechanisms to elicitability expansion, rather than characterize it; we defer a full characterization to Section 4.

### 3.1 FORMALIZING THE MECHANISMS AND MOTIVATING EXAMPLES

We formalize three natural mechanisms through which aggregation can provide benefits in our framework. For each mechanism, we illustrate through an example how the mechanism can enable an aggregation operations to expand the set of elicitable outputs. Our examples use the intersection and addition aggregation rule that we previously introduced. At the end of this subsection, we investigate the extent to which these aggregation rules can implement the mechanisms that we formalize below.

Our examples also focus on a 3-dimensional output space ($M = 3$) with 2-dimensional features ($N = 2$). We focus on feature weights matrices $\boldsymbol{\alpha}$ of the form $\boldsymbol{\alpha}_q := \begin{bmatrix} 1 & 0 & q \\ 0 & 1 & q \end{bmatrix}$. Each of the output dimensions $x_1, x_2$ specialize to features $F_1, F_2$, respectively. That is, increasing the first output dimension $x_1$ only increases the first feature $F_1$, and increasing the second output dimension $x_2$ only increases the second feature $F_2$. Increasing the third output dimension $x_3$ increases both features, though the contribution is weighted by a factor of $q$. The parameter $q$ captures the extent to which it is possible to simultaneously maximize both features.

**Mechanism 1: Feasibility Expansion.** Aggregation can help overcome the output limitations (i.e., the feasibility constraints faced by each agent), producing outputs that are outside of the feasible set. We formalize this through the following mechanism.

**Definition 3.1** (Feasibility Expansion)**.** *Given a constraint matrix $\boldsymbol{C}$, an aggregation operation $\boldsymbol{x}^{(1)}, \ldots, \boldsymbol{x}^{(K)} \rightarrow \boldsymbol{x}^{(A)}$ implements **feasibility expansion relative to $\boldsymbol{C}$** if $\boldsymbol{x}^{(A)}$ is infeasible i.e., $\boldsymbol{C}\boldsymbol{x}^{(A)} \not\leq 0$ but all $\boldsymbol{x}^{(i)}$ for $i \in [K]$ are feasible i.e., $\boldsymbol{C}\boldsymbol{x}^{(i)} \leq 0$.*

The following example illustrates how aggregation operations which implement feasibility expansion can in turn expand elicitability.

**Example 3.2.** *Let the feature map be $\boldsymbol{\alpha} = \boldsymbol{\alpha}_2$, so that increasing the third output dimension contributes significantly to both features. We view the first two output dimensions as corresponding to two types of "bad" behavior, while dimension $3$ corresponds to "good" behavior. Let $\boldsymbol{C}$ be a single constraint of the form $x_3 \leq x_1 + x_2$. The constraint captures how the model cannot produce the desirable dimension without also producing some of the undesirable dimension(s).*

*The output $[0, 0, 1]$ is outside the feasibility set since it has only desirable dimensions and hence is not elicitable with any reward specification $\beta$. The system designer can still produce this output through intersection aggregation $\boldsymbol{x}^{(1)} = [1, 0, 1], \boldsymbol{x}^{(2)} = [0, 1, 1] \rightarrow \mathcal{A}_{intersect}(\boldsymbol{x}^{(1)}, \boldsymbol{x}^{(2)}) = [0, 0, 1]$ (Proposition C.1 in Appendix C.1).*

**Mechanism 2: Overcoming Reward Specification Limitations.** Even when an output is in the feasible set, the limitations of reward specification still restrict which outputs are elicitable. Aggregation can overcome the reward specification limitations faced by the system designer, as the next two mechanisms formalize.

**Mechanism 2a: Support Expansion.** One challenge due to reward specification limitations is the impossibility of eliciting outputs with a large support.[2] Aggregation can produce combine outputs with smaller supports into an output with a larger support, as the following mechanism formalizes.

**Definition 3.3** (Support expansion). *An aggregation operation $\boldsymbol{x}^{(1)}, \ldots, \boldsymbol{x}^{(K)} \to \boldsymbol{x}^{(A)}$ implements* ***support-expansion relative to*** $i$ *if* $\mathcal{S}(\boldsymbol{x}^{(A)}) \not\subseteq \mathcal{S}(\boldsymbol{x}^{(i)})$.

Aggregation operations which implement support-expansion can in turn expand elicitability, by producing outputs with larger supports than that are elicitable by a single agent, as the following example illustrates.

**Example 3.4.** *Let the feature map be $\boldsymbol{\alpha} = \boldsymbol{\alpha}_{0.6}$. Suppose that there are no constraints $\boldsymbol{C} = \varnothing$, so elicitability challenges entirely stem from reward specification limitations. We will think of the first two dimensions as two aspects we would like our output to simultaneously capture.*

*An output vector supported on both dimensions 1 and 2 cannot be elicited directly through reward design based on $F_1$ and $F_2$ (Prop C.2 in Appendix C.2). An output supported on just one of these two dimensions can be elicited through the reward function this dimension specializes in. However, any reward focusing on both features makes dimension 3 strictly preferred over the combination of dimensions 1 and 2.*

*The system designer can still produce vector $[1/2, 1/2, 0]$ supported on both dimensions 1 and 2 through addition aggregation $\boldsymbol{x}^{(1)} = [1, 0, 0], \boldsymbol{x}^{(2)} = [0, 1, 0] \to \mathcal{A}_{add}(\boldsymbol{x}^{(1)}, \boldsymbol{x}^{(2)}; [1/2, 1/2]) = [1/2, 1/2, 0]$ (Prop C.2 in Appendix C.2).*

**Mechanism 2b: Binding Set Contraction.** The next mechanism overcomes reward specification limitations by taking advantage of the output limitations of the agent. Perhaps counterintuitively, the constraints on the output space can make it easier to elicit an output through a single reward. When a constraint is binding for an output vector, some reward-increasing directions become inaccessible to the agent, as these directions will lead to violation of the binding constraint. Aggregation can combine outputs with binding constraints into an output with fewer binding constraints.

**Definition 3.5** (Binding set contraction). *An aggregation operation $\boldsymbol{x}^{(1)}, \ldots, \boldsymbol{x}^{(K)} \to \boldsymbol{x}^{(A)}$ implements* ***binding set contraction relative to*** $i$ *if* $\mathcal{V}(\boldsymbol{x}^{(A)}) \not\supseteq \mathcal{V}(\boldsymbol{x}^{(i)})$.

Aggregation operations which implement binding set contraction can expand elicitability, as following example illustrates.

**Example 3.6.** *Let the feature map be $\boldsymbol{\alpha} = \boldsymbol{\alpha}_{0.2}$. As in the first example, we will think of $x_3$ to be a "good" dimension and $x_1, x_2$ to be "bad" dimensions. Let $\boldsymbol{C}$ be a single constraint of the form $x_1 + x_2 \leq x_3$. This constraint captures how the model cannot produce the bad dimension(s) without also producing some of the good dimension.*

*The value of $q = 0.2$ is small leading to dimension 3 being inelicitable without the constraint (Proposition C.3 in Appendix C.3). The constraint allows us to elicit a vector with some amount of $x_3$, but not a vector that has only $x_3$. The intersection aggregation operation $\boldsymbol{x}^{(1)} = [1/2, 0, 1/2], \boldsymbol{x}^{(1)} = [0, 1/2, 1/2] \to \mathcal{A}_{intersect}(\boldsymbol{x}^{(1)}, \boldsymbol{x}^{(2)}) = [0, 0, 1/2]$.*

**Implementable Mechanisms by Intersection and Addition Aggregation.** Our examples constructed problem instances that intersection aggregation can implement feasibility-expansion and binding-set contraction, while addition aggregation can implement support expansion. We turn to more general problem instances, and investigate whether each aggregation rule can implement these mechanism on any problem instance. We summarize our findings in Table 1, which shows fundamental limitations of each aggregation rule.

### 3.2 CONNECTIONS BETWEEN ELICITABILITY-EXPANSION AND MECHANISMS

Moving beyond the examples in Section 3.1, we more generally study the powers and limitations that these mechanisms provide for elicitability-expansion.

**Necessity of these mechanisms.** First, we show that if an aggregation operation expands elicitability for some feature weights matrix, it must implement at least one of the three mechanisms. Specifi-

---

[2]Kleinberg et al. (2019) studied this in single-agent environments without constraints.

cally, Theorem 3.7 shows that either the operation must implement feasibility-expansion or it must implement at least one of support-expansion or binding-set contraction for every output $x^{(i)}$.

**Theorem 3.7.** *Fix conic constraints $\boldsymbol{C}$, and any aggregation operation $\boldsymbol{x}^{(1)}, \ldots, \boldsymbol{x}^{(K)} \to \boldsymbol{x}^{(A)}$. If $\boldsymbol{x}^{(1)}, \ldots, \boldsymbol{x}^{(K)} \to \boldsymbol{x}^{(A)}$ is elicitability-expanding for some feature weights matrix $\boldsymbol{\alpha}$, then at least one of the following conditions holds:*

- *$\boldsymbol{x}^{(1)}, \ldots, \boldsymbol{x}^{(K)} \to \boldsymbol{x}^{(A)}$ is feasibility-expanding relative to $\boldsymbol{C}$ (Definition 3.1).*
- *For each $i \in [K]$, $\boldsymbol{x}^{(1)}, \ldots, \boldsymbol{x}^{(K)} \to \boldsymbol{x}^{(A)}$ is either support-expanding relative to $i$ (Definition 3.3) or binding set-contracting relative to $i$ (Definition 3.5).*

The proof of Theorem 3.7 builds on the technical tools we develop in Section 4 (i.e., Theorem 4.3).

Theorem 3.7 reveals a strong form of limitation for aggregation operations who do not implement at least one of the mechanisms (Definition 3.1, 3.5, and 3.3). Specifically, the result illustrates that if an operation does not implement the mechanisms according to the conditions in Theorem 3.7, then aggregation is not elicitability-expanding, regardless of the feature weights matrix. This result illustrates conditions under which aggregation adds no power to compound AI systems regardless of the level of prompt engineering limitations.

While these three natural mechanisms are necessary for aggregation to have power, these mechanisms are not sufficient in general. We demonstrate this and discuss some special cases where they are sufficient in Appendix B.3. In Section 4, we provide a general, necessary-and-sufficient condition that more precisely captures the power and limitations of aggregation.

## 4 CHARACTERIZING ELICITABILITY-EXPANSION IN GENERAL

In this section, we provide general characterizations of when an aggregation operation $\boldsymbol{x}^{(1)}, \ldots, \boldsymbol{x}^{(K)} \to \boldsymbol{x}^{(A)}$ is elicitability-expanding. We begin by analyzing, for a fixed feature weights matrix and feasibility constraints, whether a given aggregation operation expands elicitability (Section 4.1). We then turn to a more structural question: given only the feasibility constraints, what necessary and sufficient conditions ensure that aggregation operation is not elicitability-expanding for any feature weights matrix (Section 4.2)? These characterizations provide the technical foundation for our earlier results in Sections 3.1 and 3.2 which connected the mechanisms implemented by aggregation with elicitability-expansion.

### 4.1 CHARACTERIZING WHEN ELICITABILITY-EXPANSION SUCCEEDS

To analyze the power of aggregation, we characterize whether an aggregation operation is elicitability-expanding in a given problem-instance (i.e., given a feasibility set and feature weights). Our analysis generalizes the single-agent characterization from prior work (Kleinberg et al., 2019) to allow for output limitations (i.e., nontrivial constraints $\boldsymbol{C}$). We then leverage this characterization to analyze aggregation operations.

Given a statistic $(S, V) = (\mathcal{S}(\boldsymbol{x}), \mathcal{V}(\boldsymbol{x}))$, elicitability is determined by the structure of the set

$$\mathbb{B}_{S,V} = \underbrace{\{\boldsymbol{d} \in \mathbb{R}^M : \boldsymbol{C}_V \boldsymbol{d} \le 0\}}_{(1)} \cap \underbrace{\{\boldsymbol{d} \in \mathbb{R}^M : d_j \ge 0 \forall j \in S^c\}}_{(2)} \cap \underbrace{\{\mathbb{1}^t \boldsymbol{d} < 0\}}_{(3)}.$$

The set $\mathcal{B}_{S,V}$ captures the set of directions along which the agent can move $\boldsymbol{x}$ while maintaining the constraints $\boldsymbol{C}$ (term (1)), maintaining nonnegativity constraints (term (2)), reducing $\ell_1$ norm (term (3)). Specifically, elicitability expansion can be characterized by whether the sets $\mathcal{B}_{S,V}$ intersect with the set of feature-improving directions $\{\boldsymbol{d} \in \mathbb{R}_{\ge 0}^M \mid \boldsymbol{\alpha} \boldsymbol{d} \ge \boldsymbol{0}\}$.

**Theorem 4.1.** *Fix conic constraints $\boldsymbol{C}$, feature weights matrix $\boldsymbol{\alpha}$, and aggregation operation $\boldsymbol{x}^{(1)}, \ldots, \boldsymbol{x}^{(K)} \to \boldsymbol{x}^{(A)}$. The aggregation operation $\boldsymbol{x}^{(1)}, \ldots, \boldsymbol{x}^{(K)} \to \boldsymbol{x}^{(A)}$ is elicitability-expanding if and only if both of the following conditions hold:*

- $\mathcal{B}_{\mathcal{S}(x^{(i)}), \mathcal{V}(\boldsymbol{x}^{(i)})} \cap \{\boldsymbol{d} \in \mathbb{R}^M \mid \boldsymbol{\alpha} \boldsymbol{d} \ge \boldsymbol{0}\} = \varnothing$ *for* $i \in [K]$
- $\mathcal{B}_{\mathcal{S}(x^{(A)}), \mathcal{V}(\boldsymbol{x}^{(A)})} \cap \{\boldsymbol{d} \in \mathbb{R}^M \mid \boldsymbol{\alpha} \boldsymbol{d} \ge \boldsymbol{0}\} \ne \varnothing.$

This characterizing condition depends on both the reward specification limitation (which reflect prompt engineering limitations) via $\boldsymbol{\alpha}$ and the output limitation (which reflect model capability lim-

itations) via the conic constraints $C$. This dependence highlights the role of both forms of limitations and their interplay in determining the power of aggregation.

*Proof ideas.* The core idea is that elicitability of a vector $x$ hinges on whether the feasible-perturbation set $\mathcal{B}_{\mathcal{S}(x),\mathcal{V}(x)}$ intersects the feature-improving cone $d : \alpha d \geq 0$ (Lemmas F.3–F.4). This lets us identify when each individual $x^{(i)}$ is elicitable while the aggregate $x^{(A)}$ is not—the condition for elicitability expansion.

One direction is immediate: a nonempty intersection yields a feasible direction that strictly improves every monotone reward, certifying that $x$ cannot be elicited. The converse is subtler: if the sets are disjoint, then some reward function elicits $x$, and—as in Kleinberg et al. (2019)—it can be chosen to be linear in the features.

$\square$

### 4.2 CHARACTERIZING WHEN ELICITABILITY-EXPANSION FAILS

To analyze the limitations of aggregation, we characterize conditions under which aggregation operations are not elicitablity-expanding for *any* feature map. This represents a particularly strong form of limitation, as it rules out elicitability-expansion for all forms of reward specification limitations. The characterizing condition is stated below. We can interpret the condition as a failure of *strengthened* versions of the mechanisms. We discuss this connection to the mechanisms more in Appendix E.1.

**Definition 4.2.** *[Limitation-characterizing condition] Fix constraints $C$ and aggregation operation $x^{(1)}, \ldots, x^{(K)} \to x^{(A)}$. We say that the **limitation-characterizing condition** is satisfied for $x^{(1)}, \ldots, x^{(K)} \to x^{(A)}$ if and only if both of the following conditions are satisfied:*

1. No feasibility expansion: $x^{(1)}, \ldots, x^{(K)} \to x^{(A)}$ *doesn't implement feasibility-expansion for $C$*
2. *For all $d \in \mathcal{B}_{\mathcal{S}(x^{(A)}),\mathcal{V}(x^{(A)})}$ with $d \nleq 0$, there exists $k \in [K]$ such that both of the following two conditions holds:*
   (a) No "strengthened support expansion" for $k$: *For all $j \in \mathcal{S}(x^{(k)})^c$, $-d_j - |\mathbb{1}^t d| \leq 0$.*
   (b) No "strengthened binding-set contraction" for $k$: *For all $\gamma^{(k)} \in \mathbb{R}_{\geq 0}^{|\mathcal{V}(x^{(k)})|}$,*

$$(\gamma^{(k)})^T C_{\mathcal{V}(x^{(k)})} d - |1^t d| \cdot \left| \min_{j \in [M]} (\min(0, ((\gamma^{(k)})^T C_{\mathcal{V}(x^{(k)})})_j)) \right| \leq 0,$$

The following theorem shows that the limitation-characterizing condition is *necessary* for an aggregation operation to not expand elicitability under any feature map.

**Theorem 4.3** (Necessary). *Fix constraints $C$. If the limitation-characterizing condition is satisfied, then there does not exist a feature weights matrix $\alpha$ under which $x^{(1)}, \ldots, x^{(K)} \to x^{(A)}$ is elicitability-expanding.*

The main idea of this theorem is showing that without a strengthened version of support-expansion or binding-set contraction, an aggregation operation is bound to have no power under all feature maps. Turning to the other direction, the next theorem shows that whenever the limitation-characterizing condition is violated, the aggregation operation is not limited in the strong sense. That is, the operation expands elicitablity under *some* feature weights matrix. We prove this by constructing a feature weights matrix that makes aggregation elicitability-expanding whenever the limitation-characterizing condition does not hold.

**Theorem 4.4** (Sufficient). *Fix constraints $C$, and aggregation operation $x^{(1)}, \ldots, x^{(K)} \to x^{(A)}$. If the limitation-characterizing condition is not satisfied for $x^{(1)}, \ldots, x^{(K)} \to x^{(A)}$, then there exist feature weights $\alpha$ such that $x^{(1)}, \ldots, x^{(K)} \to x^{(A)}$ is elicitability-expanding.*

## 5 DISCUSSION

In this work, we theoretically study how aggregating multiple copies of the same model gives access to a greater set of outputs than using only a single model. Building on a principal-agent framework, our results show how aggregation must implement one of three mechanisms—feasibility-expansion,

support expansion, and binding-set contraction—in order to expand the set of elicitable outputs. Although these mechanisms are not sufficient to ensure that aggregation adds power, we show a more precise condition formed from strengthening the mechanisms is sufficient.

**Conceptual insights for system designers.** Our results offer conceptual insights into when system designers benefit from specific aggregation operations in compound AI systems. Our results characterize how the interplay between prompt engineering limitations and model capability limitations affects which types of aggregation operations are useful. Specifically, aggregation not only overcomes model capability limitations (feasibility expansion), but also overcomes prompt engineering limitations through combining multiple output characteristics (support expansion) and through taking advantage of output-level limitations (binding set-contraction). Notably, even as model capabilities continue to improve, the latter two mechanisms mean that aggregation can still be useful to system designers. On the flip side, our results illustrate how aggregation operations that do not take advantage of these mechanisms offer no power, regardless of whether the system designer employs sophisticated or unsophisticated prompt engineering practices.

**Connecting our mechanisms to empirical phenomena.** We now discuss how the mechanisms that we identify in our work—feasibility expansion, support expansion, and binding set-contraction—connect to existing empirical phenomena observed for LLMs, and could inspire directions for future empirical work. Since aggregation is only powerful when individual models are limited on their own, we begin by outlining the single-model limitations underlying each mechanism and the empirical evidence supporting them.

- The power of feasibility expansion traces back to limitations in the types of outputs that individual models can generate: specifically, when models can't exhibit certain (desirable) dimensions without exhibiting other (undesirable) dimensions as a side effect. This side effect has been empirically observed for safety versus overrefusal, where models which refuse a larger fraction of toxic outputs tend to refuse a larger fraction of safe outputs as a side effect (Cui et al., 2025). Similar side effects have been observed for alignment and hedging (Ouyang et al., 2022), and theoretically studied for creativity and factuality (Sinha et al., 2023).
- The power of support expansion traces back to challenges with eliciting outputs that perform along multiple dimensions at once in single-agent settings. This limitation has been empirically observed in cases where each dimension corresponds to a distinct user requirement. For example, prompts are often underspecified, since users may not include all of the requirements that they care about in the prompt (Yang et al., 2025). Moreover, even when users specify all their requirements, LLMs struggle to satisfy many requirements simultaneously (Wen et al., 2024; Guo et al., 2025).

We leave empirical validation of binding-set contraction—whose emergence depends on the interaction between prompt-engineering and model limitations—to future work. More broadly, since our results identify when aggregation enables these mechanisms, an important direction is to connect them to practice by testing whether real aggregation methods (e.g., debate (Du et al., 2024), prompt ensembling (Arora et al., 2023)) exhibit them. The single-model limitations discussed above suggest promising empirical settings where aggregation should add power.

**Model limitations and extensions.** Our stylized model, which builds on a classical principal-agent framework (Kleinberg et al., 2019), makes simplifying assumptions for tractability. First, while our analysis allows for nonlinear rewards $R$, we restrict the output constraints (i.e., model limitations) and the feature map (i.e., reward-specification limitations) to linear functional forms. Extending our model to allow for nonlinear limitations, which would complicate the structure of the agent's optimization program, is an interesting direction for future work. Moreover, we also assume each agent's reward depends only on its own outputs, though richer interdependencies may arise in repeated, multi-turn interactions (Du et al., 2024). Finally, while our analysis focuses on steering agents through reward design, it would be interesting to incorporate other choices, such as tool use and fine-tuning, that enable specialization in compound AI systems (BAIR Research Blog, 2024).

## 6 REPRODUCIBILITY STATEMENT

We provide full proofs of all of the results in the Appendix.

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

# A  LLM Usage Statement

We used GPT-5 and Claude Opus 4.1 to gather related work, get ideas for proofs, and to edit prose. All of the work done by LLMs was verified by the (human) authors on this paper.

# B  Additional details for Section 3

## B.1  Additional details of Section 3.1

Intersection aggregation does not implement support expansion for any problem instance, as the following result formalizes.

**Proposition B.1** (Intersection does not expand support). *Consider any aggregation operation of the form $x^{(1)}, \ldots, x^{(K)} \to x^{(A)} = \mathcal{A}_{intersect}(x^{(1)}, \ldots, x^{(K)})(x^{(1)}, \ldots, x^{(K)})$. For any $i \in [K]$, this aggregation operation does not implement support-expansion relative to $i$.*

Proposition B.1 follows from the fact that the support of $\mathcal{A}_{intersect}(x^{(1)}, \ldots, x^{(K)})$ is always a subset of the support of each $x^{(i)}$.

Intersection aggregation can implement feasibility-expansion as shown in Example 3.2 and binding set-contraction as shown in Example 3.6. In fact, these examples go one step further and demonstrate that elicitability expansion is achievable via these mechanisms.

Addition aggregation does not implement feasibility expansion for any problem instance, as the following result formalizes.

**Proposition B.2** (Addition cannot expand feasibility)**.** *Consider constraints $C$. Any aggregation operation of the form $\boldsymbol{x}^{(1)}, \ldots, \boldsymbol{x}^{(K)} \to \boldsymbol{x}^{(A)} = \mathcal{A}_{add}(\boldsymbol{x}^{(1)}, \ldots, \boldsymbol{x}^{(K)}; \boldsymbol{w})(\boldsymbol{x}^{(1)}, \ldots, \boldsymbol{x}^{(K)})$ does not implement feasibility expansion relative to $C$.*

Proposition B.2 directly follows from the fact that the constraint set $C$ is conic.

On the other hand, addition aggregation operations can implement the other two mechanisms. Example 3.4 already constructed a problem instance where addition aggregation implements support expansion. The next example constructs a problem instance where addition aggregation can implement binding set contraction (Definition 3.5) and achieve elicitability-expansion for some feature mapping.

**Example B.3** (Addition can result in binding set contraction)**.** *Consider the constraint matrix*

$$C = \begin{pmatrix} 1 & -1 & 0 \\ 1 & -\frac{1}{4} & -1 \end{pmatrix},$$

*and consider vectors $\boldsymbol{x}^{(1)} = (1, 1, 2)$ and $\boldsymbol{x}^{(2)} = (2, 4, 1)$. Note that they are both feasible and $\boldsymbol{x}^{(1)}$ is binding in the first constraint and $\boldsymbol{x}^{(2)}$ in the second. Their sum is $\mathcal{A}_{add}(\boldsymbol{x}^{(1)}, \ldots, \boldsymbol{x}^{(K)}; \boldsymbol{w})(\boldsymbol{x}^{(1)}, \boldsymbol{x}^{(2)}; [1, 1]) \to \boldsymbol{x}^{(A)} = (3, 5, 3)$, which is also feasible but does not have any binding constraints.*

## B.2 ADDITIONAL DETAILS OF SECTION 3.2

|  | **Feasibility Expansion** | **Support Expansion** | **Binding Set Contraction** |
|---|---|---|---|
| **Intersection aggregation** | ✓ (Example 3.2) | × (Proposition B.1) | ✓ (Example 3.6) |
| **Addition aggregation** | × (Proposition B.2) | ✓ (Example 3.4) | ✓ (Example B.3) |

Table 1: Implementability of mechanisms in Section 3.1 for the intersection aggregation rule equation 1 and additional aggregation rule equation 2. The symbol ✓ denotes that there exists a problem instance where the aggregation rule implements that mechanism. The symbol × denotes that the aggregation rule does not implement the mechanism for any problem instance.

**Proposition B.4.** *Fix conic constraints $C$, and any $\boldsymbol{x}^{(1)}, \ldots, \boldsymbol{x}^{(K)} \to \boldsymbol{x}^{(A)}$. Suppose that $\mathcal{V}(x^{(A)}) = \mathcal{V}(x^{(1)}) = \ldots = \mathcal{V}(x^{(K)}) = \varnothing$, and suppose that $x^{(1)}, \ldots, x^{(K)} \to x^{(A)}$ is not feasibility-expanding. Suppose also that there do **not** exist witnesses $j(i) \in \mathcal{S}(\boldsymbol{x}^{(A)}) \smallsetminus \mathcal{S}(\boldsymbol{x}^{(i)})$ for each $i \in [K]$ such that $\{j(i) \mid i \in [K]\} \neq [M]$. Then, $\boldsymbol{x}^{(1)}, \ldots, \boldsymbol{x}^{(K)} \to \boldsymbol{x}^{(A)}$ is not elicitability-expanding for any $\boldsymbol{\alpha}$.*

**Proposition B.5.** *There exists an aggregation operation $\boldsymbol{x}^{(1)}, \boldsymbol{x}^{(2)} \to \boldsymbol{x}^{(A)}$ and a set of conic constraints $C$ such that $\boldsymbol{x}^{(1)}, \ldots, \boldsymbol{x}^{(K)} \to \boldsymbol{x}^{(A)}$ implements binding-set contraction relative to $i$ for every $i \in [K]$. However, $\boldsymbol{x}^{(1)}, \boldsymbol{x}^{(2)} \to \boldsymbol{x}^{(A)}$ is not elicitability-expanding for any feature map $\boldsymbol{\alpha}$.*

**Proposition B.6.** *Fix $C = \varnothing$. There exists an aggregation operation $\boldsymbol{x}^{(1)}, \boldsymbol{x}^{(2)} \to \boldsymbol{x}^{(A)}$ such that $\boldsymbol{x}^{(1)}, \boldsymbol{x}^{(2)} \to \boldsymbol{x}^{(A)}$ implements support-expansion relative to $i$ for every $i \in [K]$. However, $\boldsymbol{x}^{(1)}, \boldsymbol{x}^{(2)} \to \boldsymbol{x}^{(A)}$ is not elicitability-expanding for any feature map $\boldsymbol{\alpha}$.*

**Proposition B.7.** *Fix conic constraints $C$ and $\boldsymbol{x}^{(1)}, \ldots, \boldsymbol{x}^{(K)} \to \boldsymbol{x}^{(A)}$. Suppose that $M \geq 2$, and $\mathcal{S}(x^{(A)}) = [M]$. Suppose that there exist witnesses $\ell(i) \in \mathcal{V}(\boldsymbol{x}^i) \smallsetminus \mathcal{V}(\boldsymbol{x}^{(A)})$ such that there exists $\boldsymbol{d}$ such that $C_{\ell(i)}\boldsymbol{d} + (|\min_{j \in [M]} C_{\ell(i),j}|) \cdot \mathbf{1}^t \boldsymbol{d} > 0$ for all $i \in [K]$, $\mathbf{1}^t \boldsymbol{d} < 0$, and $C_{\mathcal{V}(x^{(A)})}\boldsymbol{d} \leq 0$. Then, $\boldsymbol{x}^{(1)}, \ldots, \boldsymbol{x}^{(K)} \to \boldsymbol{x}^{(A)}$ is elicitability-expanding for some feature weights matrix $\boldsymbol{\alpha}$.*

## B.3 PARTIAL SUFFICIENCY OF MECHANISMS IN CONCRETE INSTANCES

In Theorem 3.7, we showed how the mechanisms are necessary for an aggregation operation to have power. We now turn to analyzing when mechanisms are sufficient for guaranteeing the power of aggregation. We focus on a weak form of power that only requires that aggregation expands

elicitability for some feature weights matrix, taking a negation of of the limitation show in Theorem 4.3. (We defer an analysis of the role of the feature weights matrix to Section 4.1.)

We first show that feasibility expansion guarantees this form of power, providing a partial converse of Theorem 3.7.

**Proposition B.8.** *Fix conic constraints* $C$. *If an aggregation operation* $\boldsymbol{x}^{(1)}, \ldots, \boldsymbol{x}^{(K)} \to \boldsymbol{x}^{(A)}$ *implements feasibility-expansion, then there exists a feature map* $\boldsymbol{\alpha}$ *such that* $\boldsymbol{x}^{(1)}, \ldots, \boldsymbol{x}^{(K)} \to \boldsymbol{x}^{(A)}$ *is elicitability-expanding.*

We now turn to support expansion and binding-set contraction. Interestingly, even if an aggregation operation implements support-expansion for every $i \in [K]$, the aggregation still may not elicitability-expanding for any feature weights matrix (Proposition B.6). Similarly, binding-set contraction also does not guarantee that aggregation has power (Proposition B.5).

Nonetheless, we show stronger conditions under which support expansion and binding-set contraction do guarantee that aggregation expands elicitability for some feature map. For support expansion, the main requirement is a global form of support expansion across outputs $i$, requiring that the "witnesses" don't span all of the output dimensions.[3]

**Proposition B.9.** *Fix conic constraints* $C$, *and any* $\boldsymbol{x}^{(1)}, \ldots, \boldsymbol{x}^{(K)} \to \boldsymbol{x}^{(A)}$. *Suppose that there exist witnesses* $j(i) \in \mathcal{S}(\boldsymbol{x}^{(A)}) \setminus \mathcal{S}(\boldsymbol{x}^{(i)})$ *for each* $i \in [K]$ *such that* $\{j(i) \mid i \in [K]\} \neq [M]$. *Suppose that* $\mathcal{V}(x^{(A)}) = \varnothing$. *Then,* $\boldsymbol{x}^{(1)}, \ldots, \boldsymbol{x}^{(K)} \to \boldsymbol{x}^{(A)}$ *is elicitability-expanding for some* $\boldsymbol{\alpha}$.

Turning to binding-set contraction, the main requirement is again a global form of binding-set contraction across outputs $i$ which links witnesses (i.e., a constraint in $\mathcal{V}(\boldsymbol{x}^{(i)}) \setminus \mathcal{V}(\boldsymbol{x}^{(A)})$ for each $i \in [K]$) together (Proposition B.7). A global variant of support expansion and binding-set contraction also emerges in our characterizations in Section 4.

# C  PROOFS FOR SECTION 3

Recall that the examples in this section use the feature weights matrix $\boldsymbol{\alpha}_q := \begin{bmatrix} 1 & 0 & q \\ 0 & 1 & q \end{bmatrix}$.

## C.1  ANALYSIS OF EXAMPLE 3.2

**Proposition C.1.** *For the feature weights matrix* $\boldsymbol{\alpha}_2$ *and constraint matrix with the row* $x_3 \leq x_1 + x_2$ *in Example 3.2,* $\boldsymbol{x}^{(1)} = [1, 0, 1], \boldsymbol{x}^{(2)} = [0, 1, 1] \to \boldsymbol{x}^{(A)} = [0, 0, 1]$ *is elicitability-expanding.*

*Proof.* From the construction, it is easy to see that $x^{(1)}$ can be elicited with a linear reward function $[1, 0, 0]$ equal to the $F_1$ and budget level $E = 2$ and $x^{(1)}$ can be elicited with a linear reward function $[0, 1, 0]$ equal to the $F_1$ and budget level $E = 2$.

Let us use our characterization Theorem 4.1 to formally elicitability-expansion.

For $\boldsymbol{x}^{(i)}$, the support set $\mathcal{S}(\boldsymbol{x}^{(i)})$ is $\{i\}$. The constraint is binding on both $x^{(i)}$. The set $\mathcal{B}_{\mathcal{S}(\boldsymbol{x}^{(1)}), \mathcal{V}(x^{(1)})}$ is $\{\boldsymbol{d} : \boldsymbol{d}_3 \leq \boldsymbol{d}_1 + \boldsymbol{d}_2, \boldsymbol{d}_2 \geq 0, \boldsymbol{d}_1 + \boldsymbol{d}_2 + \boldsymbol{d}_3 < 0\}$. For any $\boldsymbol{d}$ in this set, $\boldsymbol{d}_3 < 0$ and $\boldsymbol{d}_1 + \boldsymbol{d}_2 < -\boldsymbol{d}_3$. The set $\mathcal{B}_{\mathcal{S}(\boldsymbol{x}^{(2)}), \mathcal{V}(x^{(2)})}$ is $\{\boldsymbol{d} : \boldsymbol{d}_3 \leq \boldsymbol{d}_1 + \boldsymbol{d}_2, \boldsymbol{d}_1 \geq 0, \boldsymbol{d}_1 + \boldsymbol{d}_2 + \boldsymbol{d}_3 < 0\}$. For any $\boldsymbol{d}$ in this set, $\boldsymbol{d}_3 < 0$ and $\boldsymbol{d}_1 + \boldsymbol{d}_2 < -\boldsymbol{d}_3$.

Now consider the set of feature-improving direction $\{\boldsymbol{d} : \boldsymbol{d}_1 + 2\boldsymbol{d}_3 \geq 0, \boldsymbol{d}_2 + 2\boldsymbol{d}_3 \geq 0\}$. For any $\boldsymbol{d}$ in this set, $\boldsymbol{d}_1 + \boldsymbol{d}_2 \geq -4d_3$.

All three conditions $\boldsymbol{d}_3 < 0$, $\boldsymbol{d}_1 + \boldsymbol{d}_2 < -\boldsymbol{d}_3$, and $\boldsymbol{d}_1 + \boldsymbol{d}_2 \geq -4d_3$ cannot be satisfied since for $\boldsymbol{d}_3 < 0$, $-\boldsymbol{d}_3 > -4\boldsymbol{d}_3$. Hence there is no intersection between feasibility improving directions and features improving directions and $x^{(1)}$ is elicitable. Similarly, $x^{(2)}$ is also elicitable.

$\boldsymbol{x}^{(3)}$ is not feasible and hence not elicitable. This shows that $\boldsymbol{x}^{(1)}, \boldsymbol{x}^{(2)} \to \boldsymbol{x}^{(3)}$ is elicitability-expanding by implementing feasibility-expansion. $\qquad\square$

---

[3]The fact the witnesses cannot span all of the output dimensions condition also turns to be a necessary condition for aggregation to not be powerless (Proposition E.2).

## C.2 ANALYSIS OF EXAMPLE 3.4

**Proposition C.2.** *For the feature weights matrix $\boldsymbol{\alpha}_{0.6}$ and null constraint matrix Example 3.4, $\boldsymbol{x}^{(1)} = [1,0,0], \boldsymbol{x}^{(2)} = [0,1,0] \to \boldsymbol{x}^{(A)} = [1/2, 1/2, 0]$ is elicitability-expanding.*

*Proof.* The set of directions $\mathcal{B}_{\mathcal{S}(\boldsymbol{x}^{(1)}),\mathcal{V}(\boldsymbol{x}^{(1)})} = \{\boldsymbol{d} : \boldsymbol{d}_2 \geq 0, \boldsymbol{d}_3 \geq 0, \boldsymbol{d}_1 + \boldsymbol{d}_2 + \boldsymbol{d}_3 < 0\}$. And the set of feature-improving directions is $\mathcal{A} = \{\boldsymbol{d} : \boldsymbol{d}_1 + 0.6\boldsymbol{d}_3 \geq 0, \boldsymbol{d}_2 + 0.6\boldsymbol{d}_3 \geq 0\}$.

$\boldsymbol{d} \in \mathcal{B}_{\mathcal{S}(\boldsymbol{x}^{(1)}),\mathcal{V}(\boldsymbol{x}^{(1)})}$ means that $\boldsymbol{d}_1 < -(\boldsymbol{d}_2 + \boldsymbol{d}_3) < -\boldsymbol{d}_3$ and $\boldsymbol{d}_3 \geq 0$. $\boldsymbol{d} \in \mathcal{A}_1$ means that $\boldsymbol{d}_1 \geq -0.6\boldsymbol{d}_3$. These three conditions cannot be simultaneously showing that $\boldsymbol{x}^{(1)}$ is elicitable due to empty intersection of $\mathcal{A}$ and $\mathcal{B}_1$. Symmetrically, we can also show that $\boldsymbol{x}^{(2)}$ is also elicitable.

Now let us argue that $\boldsymbol{x}^{(A)} = [1/2, 1/2, 0]$ is not elicitable. The feasibility improving directions set is $\mathcal{B}_{\mathcal{S}(\boldsymbol{x}^{(A)}),\mathcal{V}(\boldsymbol{x}^{(A)})} = \{\boldsymbol{d} : \boldsymbol{d}_3 \geq 0, \boldsymbol{d}_1 + \boldsymbol{d}_2 + \boldsymbol{d}_3 < 0\}$. Consider $\boldsymbol{d} = [-0.6, 0.6, 1]$. $\boldsymbol{d} \in \mathcal{A} \cap \mathcal{B}_A$. This shows that $\boldsymbol{x}^{(A)}$ is not elicitable.

$\square$

## C.3 ANALYSIS OF EXAMPLE 3.6

**Proposition C.3.** *For the feature weights matrix $\boldsymbol{\alpha}_{0.2}$ and conic constraint matrix with one constraint $x_1 + x_2 \leq x_3$ from Example 3.6, $\boldsymbol{x}^{(1)} = [1,0,1], \boldsymbol{x}^{(2)} = [0,1,1] \to \boldsymbol{x}^{(A)} = [0,0,1]$ is elicitability-expanding.*

*Proof of Proposition C.3.* The feature-improving directions are the set $\mathcal{A} = \{\boldsymbol{d} : \boldsymbol{d}_1 + 0.2\boldsymbol{d}_3 \geq 0, \boldsymbol{d}_2 + 0.2\boldsymbol{d}_3 \geq 0\}$.

The constraint is binding at both $\boldsymbol{x}^{(1)}$ and $\boldsymbol{x}^{(2)}$. The feasibility improving directions are $\mathcal{B}_{(\mathcal{S}(\boldsymbol{x}^{(1)}),\mathcal{V}(\boldsymbol{x}^{(1)}))} = \{\boldsymbol{d} : \boldsymbol{d}_1 + \boldsymbol{d}_2 \leq \boldsymbol{d}_3, \boldsymbol{d}_2 \geq 0, \boldsymbol{d}_1 + \boldsymbol{d}_2 + \boldsymbol{d}_3 < 0\}$.

If $\boldsymbol{d} \in \mathcal{B}_{(\mathcal{S}(\boldsymbol{x}^{(1)}),\mathcal{V}(\boldsymbol{x}^{(1)}))}$, then $\boldsymbol{d}_1 + \boldsymbol{d}_2 + \boldsymbol{d}_3 < 0$ and $\boldsymbol{d}_1 + \boldsymbol{d}_2 + \boldsymbol{d}_3 \leq 2\boldsymbol{d}_3$. This implies that $\boldsymbol{d}_3 < 0$. If $\boldsymbol{d} \in \mathcal{A}$, then $\boldsymbol{d}_1 \geq -0.2\boldsymbol{d}_3$ and $\boldsymbol{d}_2 \geq -0.2\boldsymbol{d}_3$. If all the conditions are satisfied simultaneously, then $\boldsymbol{d}_1 > 0$ and $\boldsymbol{d}_2 > 0$. This contradicts $\boldsymbol{d}_1 + \boldsymbol{d}_2 \leq \boldsymbol{d}_3 < 0$.

The conic constraint is not binding at $\boldsymbol{x}^{(A)}$. Now consider the feasibility improving directions of $\boldsymbol{d}^{(A)}$: $\mathcal{B}_{(\mathcal{S}(\boldsymbol{x}^{(A)}),\mathcal{V}(\boldsymbol{x}^{(A)}))} = \{\boldsymbol{d} : \boldsymbol{d}_1 \geq 0, \boldsymbol{d}_2 \geq 0, \boldsymbol{d}_1 + \boldsymbol{d}_2 + \boldsymbol{d}_3 < 0\}$. The vector $\boldsymbol{d} = (0.2, 0.2, -1) \in \mathcal{A} \cap \mathcal{B}_A$ demonstrating that $\boldsymbol{x}^{(A)}$ is not elicitable. $\square$

# D PROOFS IN SECTION 3.2

## D.1 PROOF OF THEOREM 3.7

*Proof of Theorem 3.7.* We show this as a corollary of Theorem 4.3. We will prove this by showing that when both of the conditions in the theorem are violated, the limitation-characterizing condition is satisfied and hence $\boldsymbol{x}^{(1)}, \ldots, \boldsymbol{x}^{(K)} \to \boldsymbol{x}^{(A)}$ cannot be elicitability-expanding.

One of the condition of the limitation-characterizing condition is the lack of feasibility-expansion which is implied by the violation of the theorem's condition. We will show that the other condition of the limitation-characterizing condition also holds.

When the second condition of the theorem is violated, there exists $i \in [K]$ with respect to which $\boldsymbol{x}^{(1)}, \ldots, \boldsymbol{x}^{(K)} \to \boldsymbol{x}^{(A)}$ is neither support-expandin nor binding-set contracting. That is, there is an $i$ such that $\mathcal{V}(\boldsymbol{x}^{(i)}) \subseteq \mathcal{V}(\boldsymbol{x}^{(A)})$ and $\mathcal{S}(\boldsymbol{x}^{(i)}) \supseteq \mathcal{S}(\boldsymbol{x}^{(A)})$.

For every $\boldsymbol{d} \in \{C_{\mathcal{V}(\boldsymbol{x}^{(A)})}\boldsymbol{d} \leq 0, \boldsymbol{d}_{\mathcal{S}(\boldsymbol{x}^{(A)})^c} \geq 0, \mathbf{1}^\top \boldsymbol{d} = -1\}$, $C_{\mathcal{V}(\boldsymbol{x}^{(i)})}(\boldsymbol{d}) \leq 0$ and $\boldsymbol{d}_{\mathcal{S}(\boldsymbol{x}^{(i)})^c} \geq 0$, since the rows of $C_{\mathcal{V}(\boldsymbol{x}^{(i)})}$ are a subset of the rows in $C_{\mathcal{V}(\boldsymbol{x}^{(A)})}$ and similarly, the rows in $\boldsymbol{d}_{\mathcal{S}(\boldsymbol{x}^{(i)})^c} \geq 0$ are a subset of the rows in $\boldsymbol{d}_{\mathcal{S}(\boldsymbol{x}^{(A)})^c}$. Hence for any $\gamma^{(i)} \in \mathbb{R}_{\geq 0}^{|V_i|}$, $(\gamma_{\mathbf{i}})^\top C_{V_i}d - \|(\gamma_i^\top C_{V_i})_-\|_\infty \leq 0$. $\square$

## D.2 PROOF OF PROPOSITION B.8

*Proof of Proposition B.8.* This follows from Theorem 4.4. □

## D.3 PROOF OF PROPOSITION B.6

*Proof of Proposition B.6.* This follows from Proposition E.2. □

## D.4 PROOF OF PROPOSITION B.5

*Proof of Proposition B.5.* Consider a problem with two output dimensions having the following two constraints: 1) $c_1 : x_1 - x_2 \leq 0$, 2) $c_2 : -2x_1 + x_2 \leq 0$. Consider an aggregation operation $\boldsymbol{x}^{(1)} = (1/2, 1/2), \boldsymbol{x}^{(2)} = (1/2, 2/3) \rightarrow \boldsymbol{x}^{(A)} = (5/12, 7/12)$, where the binding constraints sets are $\mathcal{V}_{\boldsymbol{x}^{(i)}} = \{c_i\}$ for $i \in \{1, 2\}$ and $\mathcal{V}_{\boldsymbol{x}^{(A)}} = \varnothing$.

In this example, we will show how the limitations-characterization condition holds, meaning that the operation cannot be elicitability-expanding for any feature map $\boldsymbol{\alpha}$.

For any $\gamma_i \geq 0$ and $\boldsymbol{d}$, $\gamma_i c_i \boldsymbol{d} - \gamma_i \|(c_i)_-\|_\infty > 0$ if and only if $c_i \boldsymbol{d} - \|(c_i)_\infty\| > 0$. In this example, the existence of $\gamma_i$ for this inequality to be satisfied for each $i$ corresponds to the conditions that 1) $d_1 - d_2 > 1$ and $-2d_1 + d_2 > 2$. Since there are no elements outside the support of the vectors, there are no additional conditions to check for the limitations-characterization condition.

These two conditions imply that $1 + d_2 < d_1 < (d_2 - 2)/2$. Hence the conditions can only be satisfied when $1 + d_2 < (d_2 - 2)/2$. This is only satisfied when $d_2 < -4$ and this implies $d_1 < -3$. Hence the two conditions being satisfied means $\mathbb{1}^{d < -7}$. So the set of $\boldsymbol{d}$ such that $1\top\boldsymbol{d} = -1$ cannot intersect with the set of $\boldsymbol{d}$ satisfying the two conditions. □

## D.5 PROOF OF PROPOSITION E.2

*Proof of Proposition E.2.* Suppose that $M \geq 2$, and $\mathcal{S}(x^{(A)}) = [M]$.

We apply Theorem 4.3. It suffices to show that the limiting-characterization condition (Definition 4.2 is satisfied. By assumption, we know that the aggregation operation is not feasibility expanding. It suffices to show that that there does not exist $\boldsymbol{d}$ such that for every $i \in [K]$ there exists $j(i) \in \mathcal{S}(\boldsymbol{x}^{(i)})^c$ such that $-d_{j(i)} - |1\top\boldsymbol{d}| > 0$.

Let's show the contrapositive: assume that there exists $\boldsymbol{d}$ such that for every $i \in [K]$ there exists $j(i) \in \mathcal{S}(\boldsymbol{x}^{(i)})^c$ such that $-d_{j(i)} - |1\top\boldsymbol{d}| > 0$. Since $\boldsymbol{d} \in \mathcal{B}_{\mathcal{S}(\boldsymbol{x}^{(A)}), \mathcal{V}(\boldsymbol{x}^{(A)})}$, we know that $\boldsymbol{d}_{j'} \geq 0$ for $j' \notin \mathcal{S}(\boldsymbol{x}^{(A)})$ and we know that $1\top d < 0$. If $j \notin \mathcal{S}(\boldsymbol{x}^{(A)})$, then note that $-d_j < 0$, so this means that $j(i) \in \mathcal{S}(x^{(A)})$. Putting this together, we see that $j(i) \in \mathcal{S}(x^{(A)}) \setminus \mathcal{S}(\boldsymbol{x}^{(i)})$.

It suffices to show that $\{j(i) \mid i \in [K]\} \neq [M]$. Assume for sake of contradiction that $\{j(i) \mid i \in [K]\} = [M]$. Then since we know that $0 < -d_{j(i)} - |1\top\boldsymbol{d}| = 1\top\boldsymbol{d} - d_{j(i)}$, if we add up all of these equations in the set $\{j(i) \mid i \in [K]\}$, we would obtain that $0 < M \cdot 1\top\boldsymbol{d} - \sum_j d_j = (M - 1) \cdot 1\top\boldsymbol{d} - \sum_j d_j$, which means that $1\top\boldsymbol{d} > 0$ which is a contradiction.

□

## D.6 PROOF OF PROPOSITION B.9

*Proof of Proposition B.9.* We apply Theorem 4.4. It suffices to show that the limiting-characterization condition (Definition 4.2 is violated. Let $\boldsymbol{d}$ be the vector such that $\boldsymbol{d}_{j(i)} = -1$ for all $i \in [K]$, $\boldsymbol{d}_j = |\{j(i) \mid i \in [K]\}| - 0.5$ for some $j \notin \{j(i) \mid i \in [K]\}$, and 0 elsewhere. It follows from definition that $\boldsymbol{d} \in \mathcal{B}_{\mathcal{S}(\boldsymbol{x}^{(A)}), \mathcal{V}(\boldsymbol{x}^{(A)})}$. It suffices to show that for all $i \in [K]$, it holds that:

$$-d_{\ell(i)} - |1\top d| > 0.$$

Using that $1\top\boldsymbol{d} < 0$, this can be rewritten as:

$$-d_{\ell(i)} - |1\top d| = \sum_{j \neq \ell(i)} d_j = |\{j(i) \mid i \in [K]\}| - 0.5 - |\{j(i) \mid i \in [K]\}| + 1 = 0.5 > 0,$$

as desired.

$\square$

### D.7 Proof of Proposition B.7

*Proof of Proposition B.7.* We apply Theorem 4.4. It suffices to show that the limiting-characterization condition (Definition 4.2 is violated). For each $i \in [K]$, we take $\gamma^{(i)}$ to be the 1-hot vector with the 1 on the $\ell(i)$th condition. Let $\boldsymbol{d}$ be the vector given by the condition in the theorem statement. It follows immediately that $\boldsymbol{d} \in \mathcal{B}_{\mathcal{S}(\boldsymbol{x}^{(A)}), \mathcal{V}(\boldsymbol{x}^{(A)})}$. It suffices to show that for all $i \in [K]$, it holds that:

$$C_{\ell(i)}\boldsymbol{d} - |1^\top\boldsymbol{d}| \left| \min_{j\in[M]} \min(0, C_{\ell(i),j}) \right|.$$

Using that $1^\top\boldsymbol{d} < 0$ and using that $C_{\ell(i)}$ has at least one negative coordinate, this can be written as:

$$C_{\ell(i)}\boldsymbol{d} + 1^\top\boldsymbol{d} \left| \min_{j\in[M]} C_{\ell(i),j} \right| > 0,$$

which we know holds.

$\square$

## E Additional details for Section 4

### E.1 Connecting the limitation-characterizing condition to mechanisms

The limitation-characterization condition requires two sub-conditions to hold. The first is lack of implementation of feasibility expansion. We can interpret the second sub-condition as not implementing either a strengthening of support-expansion or a strengthening of binding-set contraction.

To demonstrate the connection between the limitation-characterizing condition and the mechanisms, we will first show that when none of the mechanisms are implemented, the limitation characterizing condition is satisfied. We will later discuss the ways in which the limitation-characterizing is related to strengthened versions of the mechanisms.

The following result shows that none of the mechanisms being implemented implies that the limitation-characterizing condition is satisfied. This result immediately implies Theorem 3.7 (i.e., that implementing at least one of these mechanisms is necessary for elicitability-expansion).

**Proposition E.1.** *Fix conic constraints $\boldsymbol{C}$, and any aggregration operation $\boldsymbol{x}^{(1)}, \ldots, \boldsymbol{x}^{(K)} \to \boldsymbol{x}^{(A)}$ where each $\boldsymbol{x}^{(k)}$ is feasible i.e., $\boldsymbol{C}\boldsymbol{x}^{(k)} \le 0$, for every $k \in [K]$.. If $\boldsymbol{x}^{(1)}, \ldots, \boldsymbol{x}^{(K)} \to \boldsymbol{x}^{(A)}$ satisfies both of the following conditions, then $\boldsymbol{x}^{(1)}, \ldots, \boldsymbol{x}^{(K)} \to \boldsymbol{x}^{(A)}$ satisfies the limitation-characterizing condition (Definition 4.2).*

- *$\boldsymbol{x}^{(1)}, \ldots, \boldsymbol{x}^{(K)} \to \boldsymbol{x}^{(A)}$ is not feasibility-expanding relative to $\boldsymbol{C}$ (Definition 3.1).*

- *There exists $k \in [K]$ such that $\boldsymbol{x}^{(1)}, \ldots, \boldsymbol{x}^{(K)} \to \boldsymbol{x}^{(A)}$ is neither support-expanding relative to $i$ (Definition 3.3) nor binding set-contracting relative to $i$ (Definition 3.5).*

*Proof.* If none of the mechanisms are implemented, then the first condition of the limitation-characterization condition, which is lack of implementation of feasibility expansion automatically holds. We will now show the second condition of the limitation condition also holds. The second condition requires two sub-conditions Condition 2a and Condition 2b in in Definition 4.2 to hold for some $k \in [K]$. We will show that each of these conditions are implemented by lack of support-expansion and lack of binding-set contraction respectively.

*No support-expansion relative to $k$ implies Condition 2a in Definition 4.2 relative to $k$.* When support-expansion is not implemented relative to $k$, all $j^{(k)} \in \mathcal{S}(\boldsymbol{x}^{(A)})$ also belongs to $\mathcal{S}(\boldsymbol{x}^{(k)})$. For all $\boldsymbol{d} \in \mathcal{B}_{\mathcal{S}(x^{(A)}), \mathcal{V}(x^{(A)})}$ and all $j^{(k)} \in \mathcal{S}(\boldsymbol{x}^{(k)})^c \subseteq \mathcal{S}(\boldsymbol{x}^{(A)})^c$, $\boldsymbol{d}_{j^{(k)}} \ge 0$. Hence $\boldsymbol{d}_{j^{(k)}} + |1^t\boldsymbol{d}|$ which is even larger than $\boldsymbol{d}_{j^{(k)}}$ is $\ge 0$ for every $j^{(k)} \in \mathcal{S}(\boldsymbol{x}^{(k)})^c$. This is the Condition 2a relative to $k$.

*No binding-set contraction relative to $k$ implies Condition 2b in Definition 4.2 relative to $k$.* No binding set contraction means that all constraints in $\mathcal{V}(\boldsymbol{x}^{(A)})$ are also in $\mathcal{V}(\boldsymbol{x}^{(k)})$. Hence every $\boldsymbol{d} \in \mathcal{B}_{\mathcal{S}(x^{(A)}), \mathcal{V}(x^{(A)})}$ satisfies all conic constraints $\ell \in \mathcal{V}(\boldsymbol{x}^{(k)})$. $\boldsymbol{d}$ also satisfies all non-negatively weighted sums of conic constraints in $\ell \in \mathcal{V}(\boldsymbol{x}^{(k)})$. Condition 2b relative to $k$ in Definition 4.2 only requires approximately satisfying the weighted sums of constraints and hence is implied by no binding-set contraction relative to $k$. $\square$

**How the limitation-characterizing condition strengthens mechanisms the mechanisms.** Next, we will describe how the limitation-characterizing condition, specifically Conditions 2a,2b in Definition 4.2 are failures of strictly strengthened versions of the mechanisms. This makes the limitation-characterizing condition a strictly weaker condition to be satisfied compared to failure of all mechanisms. The conditions 2a,2b of the limitation-characterizing condition are strengthenings in two ways. The first is due to requiring violations by minimum margins. Note that from the proof of Proposition E.1, just violation without any minimum margin requirement suffices for the mechanisms. Another way that these conditions are stronger is the *joint* requirement across all $k \in [K]$. We require that the same $\boldsymbol{d} \in \mathcal{B}_{\mathcal{S}(\boldsymbol{x}^{(A)}), \mathcal{V}(\boldsymbol{x}^{(A)})}$ witnesses the violation by a margin for every $k \in [K]$.

In some special cases, the limitation-characterizing conditions correspond exactly to not implementing any of the mechanisms, instead of not implementing strengthenings. One special case is when no vector in the aggregation operation has any binding conic constraints. This holds when there are no conic constraints. In this special case, even the regular, non-strengthened form of binding-set contraction cannot kick in. We can show that in this special case, the limitation-characterizing condition is either not feasibility-expansion or not the usual, non-strengthened support-expansion as long as a particular edge case does not occur.

**Corollary E.2.** *Fix conic constraints $\boldsymbol{C}$, and any $\boldsymbol{x}^{(1)}, \ldots, \boldsymbol{x}^{(K)} \to \boldsymbol{x}^{(A)}$. Suppose that $\mathcal{V}(x^{(A)}) = \mathcal{V}(x^{(1)}) = \ldots = \mathcal{V}(x^{(K)}) = \varnothing$, and suppose that $x^{(1)}, \ldots, x^{(K)} \to x^{(A)}$ is not feasibility-expanding. Then $\boldsymbol{x}^{(1)}, \ldots, \boldsymbol{x}^{(K)} \to \boldsymbol{x}^{(A)}$ is elicitability-expanding for some $\boldsymbol{\alpha}$ if and only if (1) if $x^{(A)}$ is full-support i.e., $\mathcal{S}(\boldsymbol{x}^{(A)}) = [M]$, then there exists $j \in [M]$ such that so $\boldsymbol{x}^{(k)}$ has support $[M] \setminus \{j\}$ **and** (2) $\boldsymbol{x}^{(A)}$ is support-expanding relative to every $k \in [K]$ i.e., $\mathcal{S}(\boldsymbol{x}^{(A)}) \nsubseteq \mathcal{S}(\boldsymbol{x}^{(k)})$ for every $k \in [K]$.*

It is harder to remove the strengthening for the binding set constraints. This is due to the joint geometry of the constraints that appears in the limitation-characterizing condition.

# F    Proofs for Section 4

## F.1    Key Lemmas for Section 4

The following lemmas provides the characterization for the elicitability of a vector $\boldsymbol{x}$ under a feature weights matrix $\boldsymbol{\alpha}$ in terms of the intersection of feasible perturbation directions $\mathcal{B}_{\mathcal{S}(\boldsymbol{x}), \mathcal{V}(\boldsymbol{x})} = \{\boldsymbol{d} : \boldsymbol{C}_{\mathcal{V}(\boldsymbol{x})} \boldsymbol{d} \leq 0\} \cap \{\boldsymbol{d} : \boldsymbol{d}_{\mathcal{S}(\boldsymbol{x})^c} \geq 0\} \cap \{1^t \boldsymbol{d} < 0\}$ and feature-improving directions $d \in \mathbb{R}^M : \{\boldsymbol{d} : \boldsymbol{\alpha} \boldsymbol{d} \geq 0\}$. These results generalize the characterization results in Kleinberg et al. (2019) to allow for conic constraints $\boldsymbol{C}$.

**Lemma F.1.** *If a vector $\boldsymbol{x}$ is elicitable with budget $E$, then $\|\boldsymbol{x}\|_1 = E$.*

*Proof.* This is because, for any feasible vector $\boldsymbol{x}$ with $\|\boldsymbol{x}\|_1 < E$, scaling $\boldsymbol{x}$ to obtain $\boldsymbol{x}' = E\boldsymbol{x}/\|\boldsymbol{x}\|_1$ results in a feasible vector that has strictly larger reward for any reward function.

$\boldsymbol{x}'$ clearly maintains nonnegativity constraints and bounded $\ell_1$ norm constraint. Additionally since the only other constraints are conic, scaling the feasible $\boldsymbol{x}$ non-negatively also maintains the additional conic constraint.

By the monotonicity of the reward functions we consider, for all reward functions, $\boldsymbol{x}'$ has reward at least as high as $\boldsymbol{x}$.

By the strict monontonicity of our feature mapping functions and for the notion of monotonicity of reward functions we consider, $\boldsymbol{x}'$ achieves a strictly higher reward than $\boldsymbol{x}$.

$\square$

The following lemma shows that elicitability of a vector only depends on the direction of the vector and not of the norm. It allows us to study elicitability of the normalized vector using budget 1 i.e., $\ell_1$ norm bound of one. Hence our elicitability characterizations will be expressed with budget 1.

**Lemma F.2.** *A vector $\boldsymbol{x}$ is elicitable with some budget $E$, under a reward function $R$ if and only if $\boldsymbol{x}/\|\boldsymbol{x}\|_1$ is elicitable with budget 1 for the same reward function.*

*Proof.* If $\boldsymbol{x}$ is elicitable, it is elicitable with a budget of $\|\boldsymbol{x}\|_1$ by Lemma F.1. It is elicitable if and only there is no feasible $\boldsymbol{y}$ with $\|\boldsymbol{y}\|_1 \leq \|\boldsymbol{x}\|_1$ with higher reward tham $\boldsymbol{x}$. If such a $\boldsymbol{y}$ exists, then $\boldsymbol{x}/\|\boldsymbol{x}\|_1$ is not elicitable with budget 1 since $\boldsymbol{y}/\|\boldsymbol{x}\|_1$ also has budget 1, is feasible and has higher reward than $\boldsymbol{x}$. Similarly, if an improving $\boldsymbol{y}$ existed for $\boldsymbol{x}/\|\boldsymbol{x}\|_1$ under budget 1, then $\boldsymbol{y}\|\boldsymbol{x}\|_1$ is improving for $\boldsymbol{x}$ under budget $\|\boldsymbol{x}\|_1$. □

**Lemma F.3** (Single output elicitation necessary). *An output vector $\boldsymbol{x}$ is elicitable only if $\mathcal{B}_{\mathcal{S}(\boldsymbol{x}),\mathcal{V}(\boldsymbol{x})} \cap \{\boldsymbol{\alpha}\boldsymbol{d} \geq 0\}$ is non-empty.*

*Proof of Lemma F.3.* Let $\boldsymbol{d} \in \mathcal{B}_{\mathcal{S}(\boldsymbol{x}),\mathcal{V}(\boldsymbol{x})} \cap \{\boldsymbol{\alpha}\boldsymbol{d} \geq 0\}$. It suffices to construct a feasible output vector $\boldsymbol{y}$ that has strictly higher reward than $\boldsymbol{x}$ for every $\boldsymbol{x}$ with $\ell_1$ norm equal to one and for every monotone reward function of the features. This is sufficient to prove the lemma since by lemma F.1, any elicitable vector has $\ell_1$ norm equal to one.

This vector $\boldsymbol{y}$ we construct is $\boldsymbol{y} = (\boldsymbol{x} + \lambda\boldsymbol{d})/\|\boldsymbol{x} + \lambda\boldsymbol{d}\|_1$ where $\lambda > 0$ is chosen to be small enough so that $\boldsymbol{y} \geq 0$.

First consider the vector $\boldsymbol{y}' = \boldsymbol{x} + \lambda\boldsymbol{d}$ for an appropriate choice of $\lambda > 0$ that we will describe in a bit. First note that $\boldsymbol{y}'$ is feasible on all conic and non-negativity constraints that are binding at $\boldsymbol{x}(x)$ due to $\boldsymbol{d}$'s membership in $\{\boldsymbol{d} : C_{\mathcal{V}(\boldsymbol{x})}\boldsymbol{d} \leq 0\} \cap \{\boldsymbol{d} : \boldsymbol{d}_{\mathcal{S}(\boldsymbol{x})^c} \geq 0\}$.

We can choose $\lambda$ to be small enough so that $\boldsymbol{y}'$ continues to meet all non-binding constraints. That is choose $\lambda < \min_{j\in\mathcal{V}(\boldsymbol{x})^c,C_j\boldsymbol{d}>0} -\mathbf{C}_j\boldsymbol{x}/\mathbf{C}_j\boldsymbol{d}$ and $\min_{i\in(\boldsymbol{x}),\boldsymbol{d}_i<0} -\boldsymbol{x}_i/\boldsymbol{d}_i$. This establishes that we have a positive choice of $\lambda$ making $\boldsymbol{y}'$ satisfy the nonnegativity and conic constraints. Additionally, we have that $\mathbb{1}^t\boldsymbol{y}' = \|\boldsymbol{y}'\|_1 = \|\boldsymbol{x}\|_1 - \lambda\mathbb{1}^t d < \|\boldsymbol{x}\|_1 = 1$. That is, $\boldsymbol{y}'$ satisfies the bounded $\ell_1$ norm constraint in a non-binding manner. This shows that $\boldsymbol{y}'$ is feasible.

We also have that $\boldsymbol{\alpha}^t\boldsymbol{y}' = \boldsymbol{\alpha}^t(\boldsymbol{x} + \boldsymbol{d}) \geq \boldsymbol{\alpha}^t\boldsymbol{x}$ since $\boldsymbol{\alpha}^t\boldsymbol{d} \geq 0$. Hence $\boldsymbol{y}'$ satisfies feasibility constraints and has at least as high values on all features. By the monotonicity of the reward functions we consider, for all reward functions, $\boldsymbol{y}'$ has reward at least as high as $\boldsymbol{x}$. Lemma F.1 shows that scaling $\boldsymbol{y}'$ to have $\ell_1$ norm equal to one results in strictly higher reward for all reward functions. Hence $\boldsymbol{y}'/\|\boldsymbol{y}'\|_1$ is feasible and has strictly higher reward than $\boldsymbol{x}$ for all monotone reward functions.

□

**Lemma F.4** (Single output elicitation sufficient). *An output vector $\boldsymbol{x}$ is elicitable if $\mathcal{B}_{\mathcal{S}(\boldsymbol{x}),\mathcal{V}(\boldsymbol{x})} \cap \{\boldsymbol{\alpha}\boldsymbol{d} \geq 0\}$ is non-empty.*

*Proof.* Write $S := S(x)$ and $V := V(x)$.

**Existence of multipliers.** By positive scaling of directions, the assumption $B_{S,V} \cap D_\alpha = \varnothing$ is equivalent to infeasibility of the system :

$$C_V d \leq 0, \qquad d_{S^c} \geq 0, \qquad \alpha^\top d \geq 0, \qquad \mathbf{1}^\top d < 0. \tag{3}$$

Let $I_{S^c} \in \mathbb{R}^{|S^c|\times M}$ be the coordinate-selector matrix whose rows are the vectors $e_j^\top$ for $j \in S^c$, so that $I_{S^c}d = d_{S^c}$.

By Motzkin's transposition theorem of the alternative, infeasibility of equation 3 implies the existence of multipliers (i.e., dual variables)

$$\gamma \in \mathbb{R}_{\geq 0}^{|V|}, \quad \lambda \in \mathbb{R}_{\geq 0}^{|S^c|}, \quad \nu \in \mathbb{R}_{\geq 0}^N, \quad \tau > 0$$

such that

$$C_V^\top \gamma - I_{S^c}^\top \lambda + \tau - \alpha^\top \nu = 0 \tag{4}$$

holds. (The strict right-hand side $\mathbf{1}^\top d < 0$ yields $\tau > 0$.)

**Reward function construction.** Define a reward function that is linear in the features

$$R(z) = \sum_{i=1}^{N} \beta_i z_i \qquad \text{with} \qquad \beta_i := \frac{\nu_i}{f_i'\big((\alpha^\top x)_i\big)} \quad (> 0),$$

which is well-defined since each $f_i$ is strictly increasing, hence $f_i'((\alpha^\top x)_i) > 0$. Let $r(u) :=$ $R(F(u)) = \sum_{i=1}^{N} \beta_i f_i\big((\alpha^\top u)_i\big)$. Because each $f_i$ is concave and increasing, $r$ is concave. Its gradient at $x$ is

$$\nabla r(x) = \sum_{i=1}^{N} \beta_i f_i'\big((\alpha^\top x)_i\big) \alpha_{\cdot,i} = \alpha \nu,$$

where $\alpha_{\cdot,i}$ is the $i$-th column of $\alpha$.

**Elicitability.** Consider the reward maximization program

$$\max_{u \in \mathbb{R}^M} r(u) \quad \text{s.t.} \quad Cu \le 0, \ u \ge 0, \ \mathbf{1}^\top u \le 1.$$

This is a concave program, and its Lagrangian is

$$\mathcal{L}(u, \lambda_0, \mu, \tilde{\gamma}) = r(u) + \lambda_0 \left(1 - \mathbf{1}^\top u\right) + \mu^\top u - \tilde{\gamma}^\top (Cu),$$

with multipliers $\lambda_0 \ge 0$, $\mu \ge 0$, $\tilde{\gamma} \ge 0$. Evaluate the KKT conditions at $u = x$ with the choice

$$\lambda_0 := \tau, \qquad \mu_S := 0, \ \mu_{S^c} := \lambda, \qquad \tilde{\gamma}_V := \gamma, \ \tilde{\gamma}_{V^c} := 0.$$

Primal feasibility holds by definition of $S, V$. Complementary slackness holds since $x_j = 0$ for $j \in S^c$ and $(Cx)_\ell = 0$ for $\ell \in V$, while $\mu_S = 0$. For stationarity,

$$\nabla r(x) - \lambda_0 \mathbf{1} + \mu - C^\top \tilde{\gamma} = \alpha g - \tau \mathbf{1} + I_{S^c}^\top \lambda - C_V^\top \gamma = 0$$

by equation 4. Finally, $\lambda_0 = \tau > 0$ certifies that the $\ell_1$-budget binds ($\mathbf{1}^\top x = 1$, consistent with Lemma C.2).

Since $r$ is concave and the constraints are linear, the KKT conditions are sufficient; hence $x$ maximizes $r$ over the feasible region and is therefore elicitable. $\qquad\square$

## F.2   Proof of Theorem 4.1

Theorem 4.1 follows directly from the single-agent results in the previous subsection.

*Proof of Theorem 4.1.* We apply Lemma F.3 and Lemma F.4 to obtain necessary and sufficient conditions on when $\boldsymbol{x}$ is elicitable. We apply this to the outputs $\boldsymbol{x}^{(1)}, \dots, \boldsymbol{x}^{(K)}$ as well as $\boldsymbol{x}^{(A)}$. $\quad\square$

## F.3   Key Intermediate Results for the Proof of Theorem 4.3 and Theorem 4.4

To prove Theorem 4.3 and Theorem 4.4, we will use an alternate but equivalent way of expressing the limitations-characterizing condition (Definition 4.2). This equivalent condition is defined below.

**Definition F.5.** *Fix constraints $C$ and aggregation operation $\boldsymbol{x}^{(1)}, \dots, \boldsymbol{x}^{(K)} \to \boldsymbol{x}^{(A)}$. We say that the alternate limitations-characterizing condition is satisfied for $\boldsymbol{x}^{(1)}, \dots, \boldsymbol{x}^{(K)} \to \boldsymbol{x}^{(A)}$ if 1) $\boldsymbol{x}^{(1)}, \dots, \boldsymbol{x}^{(K)} \to \boldsymbol{x}^{(A)}$ does not implement feasibility-expansion, and 2) there does not exist $\boldsymbol{d}^{(A)} \in \mathcal{B}_{\mathcal{S}(x^{(A)}), \mathcal{V}(x^{(A)})}$ such that:*

$$\left\{ \boldsymbol{u} + \lambda \boldsymbol{d}^{(A)} \mid \boldsymbol{u} \in \mathbb{R}_{\ge 0}^M, \lambda \ge 0 \right\} \bigcap \left( \bigcup_{i \in [k]} \mathcal{B}_{\mathcal{S}(x^{(i)}), \mathcal{V}(x^{(i)})} \right) = \varnothing.$$

The following proposition shows that the limitation-characterizing condition is equivalent to the new condition we defined above.

**Proposition F.6.** *The conditions defined in Definition 4.2 and Definition F.5 are equivalent.*

*Proof.* For ease of notation, let $V_i =: \mathcal{V}(x^{(i)})$ for $i \in [K]$ and let $V_A := \mathcal{V}(x^{(A)})$. It suffices to show that $\{u + \lambda d : u, \lambda \geq 0\}$ for a $d \in \mathcal{B}_{[M], \mathcal{V}_{\boldsymbol{x}^{(0)}}}$ has empty intersection with $\mathcal{B}_{[M], \mathcal{V}_{\boldsymbol{x}^{(i)}}}$, for each $i \in [K]$ if and only if for every $\gamma^{(i)} \in \mathbb{R}^{|V_i|}_{\geq 0}$, $\gamma_i^T C_{V_i} d - \|(\gamma_i^T C_{V_i})_-\|_\infty > 0$ or $I_{S_A^c} \boldsymbol{d} < \mathbb{1}^d \boldsymbol{d}$. Without loss of generality, it suffices to prove this for all $\gamma^{(i)} \in \mathbb{R}^{|V_i|}_{\geq 0}$ with bounded norm, say $\|\gamma^{(i)}\|_1 \leq 1$.

For any $d \in \mathcal{B}_{[M], \mathcal{V}_{\boldsymbol{x}^{(0}}}$, the intersection of $\{u + \lambda d : u, \lambda \geq 0\}$ and $\mathcal{B}_{[M], \mathcal{V}_{\boldsymbol{x}^{(i}}}$ is non-empty if and only if there exists a $u, \lambda \geq 0$ such that $C_{V_i}(u + \lambda d) \leq 0$ and $\mathbb{1}^t(u + \lambda d) < 0$.

$d \in \mathcal{B}_{[M], \mathcal{V}_{\boldsymbol{x}^{(0}}}$ means that $\mathbb{1}^t d < 0$, $C_{V_A} d \leq 0$, and $-I_{S_A^c} d \leq 0$. We can always normalize $d$ so that $\mathbb{1}^t d = -1$. We can also scale the inequalities for non-empty intersection by dividing by $\lambda$. (Note that $\lambda \neq 0$, since $1^t u \geq 0$.) Hence, we can equivalently write the condition for non-empty intersection as the existence of $v \geq 0$ such that $C_{V_i}(d + v) \leq 0$, $-I_{S_i^c}(d + v) \leq 0$ and $\mathbb{1}^t v < -\mathbb{1}^d d = 1$. These inequalities for the non-empty intersection condition hold if and only if all weighted sums (with non-negative weights) of the inequalities also hold true. That is, for every $\gamma^{(i)} \geq 0, \lambda^{(i)} \geq 0$, weight vectors, $\gamma^{(i)t} C_{V_i}(d + v) - \lambda^{(i)\top} I_{S_i^c}(d + v) \leq 0$ and $\mathbb{1}^t v < 1$.

A $v$ satisfying $(\gamma^{(i)t} C_{V_i} - \lambda^{(i)\top} I_{S_i^c})(d + v) \leq 0$ and $\mathbb{1}^t v < 0$ to simultaneously exists if and only if

$$\inf_{v \geq 0 : \mathbb{1}^t v \leq 1} \sup_{\gamma^{(i)} \geq 0, \|\gamma^{(i)}\|_1 \leq 1} (\gamma^{(i)t} C_{V_i} - \lambda^{(i)\top} I_{S_i^c})(d + v) \leq 0.$$

Since $(\gamma^{(i)t} C_{V_i} - -\lambda^{(i)\top} I_{S_i^c})(d + v)$ is an affine function in $\gamma^{(i)}, \lambda^{(i)}$ and $v$, and since the sets we optimize over $\{\boldsymbol{\gamma}^{(i)} \geq 0, \|\boldsymbol{\gamma}^{(i)}\|_1 \leq 1\}$ and $\{\boldsymbol{v} \geq 0, \mathbb{1}^t \boldsymbol{v} \leq 1\}$ are convex and compact, we can apply, we can apply minimax theorem to get

$$\inf_{v \geq 0 : \mathbb{1}^v < 1} \sup_{\gamma^{(i)}, \lambda^{(i)} \geq 0, \|\gamma^{(i)}\|_1 \leq 1, \|\lambda^{(i)}\|_1 \leq 1} (\gamma^{(i)t} C_{V_i} - \lambda^{(i)\top} I_{S_i^c})(d + v)$$

$$= \sup_{\gamma^{(i)}, \lambda^{(i)} \geq 0, \|\gamma^{(i)}\|_1 \leq 1, \|\lambda^{(i)}\|_1 \leq 1} \inf_{v \geq 0 : \mathbb{1}^v < 1} (\gamma^{(i)t} C_{V_i} - \lambda^{(i)\top} I_{S_i^c})(d + v).$$

Note that for a given $\boldsymbol{\gamma}^{(i)}, \lambda^{(i)}$, we can construct an optimal $\boldsymbol{v}$ as follows. If $\gamma^{(i)t} C_{V_i} - \lambda^{(i)\top} I_{S_i^c}$ has a negative coordinate, then $\boldsymbol{v}$ places a weight of $1$ on the most negative coordinate of $\gamma^{(i)t} C_{V_i} - \lambda^{(i)\top} I_{S_i^c}$. Otherwise, then $\boldsymbol{v} = \boldsymbol{0}$. Using this construction, we know that:

$$\inf_{v \geq 0 : \mathbb{1}^v \leq 1} (\gamma^{(i)t} C_{V_i} - \lambda^{(i)\top} I_{S_i^c})(d + v) = (\gamma^{(i)t} C_{V_i} - \lambda^{(i)\top} I_{S_i^c})d - \|(\gamma^{(i)t} C_{V_i} - \lambda^{(i)\top} I_{S_i^c})_-\|_\infty.$$

Thus, the condition of non-empty intersection becomes the condition that $(\gamma^{(i)t} C_{V_i} - \lambda^{(i)\top} I_{S_i^c})d - \|(\gamma^{(i)t} C_{V_i} - \lambda^{(i)\top} I_{S_i^c})_-\|_\infty \leq 0$ for all $\gamma^{(i)}, \lambda^{(i)} \geq \boldsymbol{0}, \|\boldsymbol{\gamma}\|_1, \|\boldsymbol{\lambda}\|_1 \leq 1$.

Note that $\gamma^{(i)t} C_{V_i} - \lambda^{(i)\top} I_{S_i^c}$ subtracts $\lambda_j$ from some coefficient of the $j$th row of $\gamma^{(i)t} C_{V_i}$. As a result, we can write $\|(\gamma^{(i)t} C_{V_i} - \lambda^{(i)\top} I_{S_i^c})_-\|_\infty$ as $\|\gamma^{(i)t} C_{V_i -}\|_\infty + \|\lambda^{(i)\top} I_{S_i^c -}\|_\infty = \|\gamma^{(i)t} C_{V_i -}\|_\infty + 1$.

So the condition $(\gamma^{(i)t} C_{V_i} - \lambda^{(i)\top} I_{S_i^c})d - (\|\gamma^{(i)t} C_{V_i -}\|_\infty + 1) \leq 0$ is equivalent to the condition that $\gamma^{(i)t} C_{V_i} - \|\gamma^{(i)t} C_{V_i -}\|_\infty \leq 0$ and $-\lambda^{(i)\top} d - 1 \leq 0$ (since both terms being $\leq 0$ implies the sum is $\leq 0$ and conversely, if the sum is not $\leq 0$, one must be $> 0$). This is exactly the condition in the limitation-characterizing condition.

$\square$

### F.4 PROOF OF THEOREM 4.3

Using this equivalence, we will show the necessity of the alternative condition to establish the necessity of the limitations-characterizing condition

*Proof of Theorem 4.3.* We will prove the contrapositive: If $\boldsymbol{x}^{(1)}, \ldots, \boldsymbol{x}^{(K)} \to \boldsymbol{x}^{(A)}$ is elicitability-expanding for some feature map $\boldsymbol{\alpha}$ and for conic constraints $\boldsymbol{C}$, then the limitations-characterizing condition (Definition 4.2) is violated.

One case is that $x^{(1)}, \ldots, x^{(K)} \to x^{(A)}$ is elicitability-expanding through feasibility-expansion. This automatically violates the limitations-characterizing condition.

The other case is that $x^{(1)}, \ldots, x^{(K)} \to x^{(A)}$ is not feasibility-expanding. Then $x^{(1)}, \ldots, x^{(K)} \to x^{(A)}$ is feasible. We will show that if the limitations-characterizing condition

If the violation occurs through existence of $x^{(1)}, \ldots, x^{(K)}$ is not elicitable.

Suppose that $x^{(A)}$ is not elicitable under a feature mapping $\alpha$ and constraints $C$. We will show that a violation of the limitations-characterizing condition implies that one of $x^{(1)}, \ldots, x^{(k)}$ is not elicitable, which contradicts $x^{(1)}, \ldots, x^{(K)} \to x^{(A)}$ being elictability-expanding.

Since $x^{(A)}$ is not elicitable under a feature weights matrix $\alpha$, by Lemma F.4, there is a $d^{(A)} \in \mathcal{K}_{S_0, V_0}$ such that $\alpha d^{(A)} \geq 0$. Since the limitation-characterizing condition is violated, the alternate limitation-characterizing condition is also violated (Proposition F.6). This means that there exists $x^{(i)}$ with $\mathcal{K}_{S_i, V_i}$ having non-empty intersection with $\{u + \lambda d^{(0)}\}$.

It suffices to show that $x^{(i)}$ is not elicitable under feature mapping $\alpha$. To see this, let $d_i$ denote an element of the intersection $\mathcal{K}_{S_i, V_i} \cap \{u + \lambda d^{(0)}\}$. We can then write $d_i = u + \lambda d^{(A)}$. Note that $\alpha d_i = \alpha u + \lambda \alpha d^{(A)}$. We know that $\alpha u \geq 0$ since $u \geq 0$ and $\alpha$ has non-negative entries. Additionally, $\alpha d^{(A)} \geq 0$ as shown above. Hence $\alpha d_i \geq 0$. By Lemma F.3, this means that $x_i$ is not elicitable.

$\square$

## F.5 Proof of Theorem 4.4

*Proof of Theorem 4.4.* Suppose the limitation-characterizing condition is satisfied. By Proposition F.6, this means that the alternate limitation-characterizing condition is satisfied. Then we know that we are in one of two cases.

**Case 1:** $x^{(1)}, \ldots, x^{(K)} \to x^{(A)}$ **implements feasibility expansion.** Consider a feature mapping with a single feature and all dimensions contribute equal weights of one to this feature. All output vectors with the same $\ell_1$ norm result in the same reward for all reward functions, and thus all feasible outcomes are elicitable. That is any output vector is elicitable if and only if it is feasible. Under this construction, feasibility-expansion implies elicitability-expansion.

**Case 2: there exists** $d^{(A)} \in \mathcal{K}_{S(x^{(A)}), \mathcal{V}(x^{(A)})}$ **such that for all** $u \geq 0, \lambda \geq 0$, $u + \lambda d^{(A)} \notin \mathcal{K}_{S_i, V_i}$ **for** $i \neq 0$**.** We will construct a feature mapping $\alpha$ based on $d^{(A)}$ such that the set of directions weakly increasing feature values i.e., the set $D_\alpha = \{d : \alpha d \geq 0\}$ is a subset of $\{u + \lambda d^{(A)} : u \geq 0, \lambda \geq 0\}$. This implies that for all other outputs $x_i$, $D_\alpha \cap \mathcal{K}_{S(x^{(i)}), \mathcal{V}(x^{(i)})}$ is empty and hence $x^{(i)}$ is elicitable under $\alpha$.

To complete this argument, we will explicitly construct such an $\alpha$ based on $d^{(A)}$. Let $P_0 = \{i \in [m] : d_i^{(A)} > 0\}$ denote the positive coordinates of $d^{(A)}$ and let $N_0 = \{i \in [m] : d_i^{(A)} \leq 0\}$ denote the negative or zero coordinates. We construct two sets of features:

- For every $p \in P_0$, there is a corresponding feature $F_p$ whose row in $A$ is the vector $e_p$ which is the vector with 1 at coordinate $p$ and zero everywhere else. That is, the action $x_p$ has weight 1 on feature $F_p$ and all other actions have zero weight.

- The next set of features are defined for every pair $p \in P_0$, $q \in N_0$. This feature $F_{p,q}$ has a corresponding row in $A_0$ that is the vector $d_p^{(A)} e_q - d_q^{(A)} e_p$. That is, the only actions with possible non-zero weights to $F_{p,q}$ are actions $x_p, x_q$. The weight from $x_p$ is $|d_q^{(A)}|$ and the weight from $x_q$ is $|d_p^{(A)}|$.

Now let us show that the set $D_\alpha = \{d : \alpha d \geq 0\}$ is a subset of $B_0 = \{u + \lambda d^{(A)}\}$. Take any $d \in D_\alpha$.

For every $p \in P_0$, since $d$ weakly improves value of $F_p$, it holds that $d_p \geq 0$. By ensuring that $\lambda \leq d_p / d_p^{(A)}$ for all $p \in P_0$, we can ensure that $d_p - \lambda d_p^{(A)} \geq 0$.

For every $p \in P_0, q \in N_0$, since $d$ weakly improves value of $F_{p,q}$, it holds that $-d_p d_q^{(A)} + d_q d_p^{(A)} \geq 0$. In other words, $d_q \geq d_p d_q^{(A)} / d_p^{(A)}$.

We will show that it is possible to choose a $\lambda \geq 0$ such that $d - \lambda d^{(A)} \geq 0$, and hence $d$ can be expressed as $u + \lambda d^{(A)}$ for $u \geq 0$. If there is a $p \in P_0$ with $d_p = 0$, then $d_q \geq 0$ while $d_q^{(A)} \leq 0$. So for all $\lambda > 0$, $d_q - \lambda d_q^{(A)} \geq 0$. Otherwise, we can choose $\lambda$ less than $d_p / d_p^{(A)}$ and we get $d_q - \lambda d_q^{(A)} \geq 0$.

$\square$

## G   EMPIRICAL SETUP FOR SECTION 2.4

**Model output generation.** The outputs are generated using gpt-4o-mini-2024-07-18 with the temperature set to $1.0$. These are the five prompts that are used to produce model outputs:

1. "From a machine learning theory perspective, list 10 influential papers that have shaped our current understanding of large language models."

2. "From the perspective of natural language processing and computational linguistics, list 10 key research papers that have been most influential in the development of modern large language models."

3. "From a cognitive science and psycholinguistics standpoint, list 10 important papers that inform our understanding of how large language models represent, process, or acquire linguistic and conceptual structure."

4. "From the standpoint of AI alignment and human–AI interaction, list 10 important papers that have shaped how large language models are aligned, instructed, or trained with feedback."

5. "From a multi-agent and game-theoretic perspective, list 10 influential papers that contribute to the development or understanding of large language models"

These prompts produce five outputs $X_1, \ldots, X_5$, each a list of 10 papers tailored to its respective perspective. Next, we pass the concatenated outputs $(X_1, \ldots, X_5)$ to gpt-4o-mini-2024-07-18 by prompting the model with *aggregation instructions* followed by the concatenation of the 5 lists of papers, where each list is preceded by followed by "List of papers: [insert output number]". The intersection-style and addition-style aggregation operations are performed using the following *aggregation instructions*.

- *Addition-style aggregation:* "Each of the following lists contains influential papers on large language models in specializaing in different areas: machine learning theory, natural language processing, computational linguistics, AI alignment, human–AI interaction, and multi-agent systems. Based on these lists, generate a new list of 10 papers that reflects the union of their themes and coverage. Your list should be freshly generated (not a literal set union), but it should include papers that plausibly come from any of the provided lists, covering as much of the combined topical space as possible."

- *Intersection-style aggregation:* "Each of the following lists contains influential papers on large language models in specializaing in different areas: machine learning theory, natural language processing, computational linguistics, AI alignment, human–AI interaction, and multi-agent systems. Based on these lists, generate a new list of 10 papers that reflects their intersection. That is, papers belonging to many of these areas of specialization. Your list should be freshly generated (not a literal intersection), selecting papers that could plausibly appear in all of the lists. If the literal intersection is empty, still generate the best possible list of papers that are central, broadly relevant, and thematically compatible with all lists."

These aggregation prompts produce outputs $X_{\text{addition}}, X_{\text{intersection}}$.

**Output vector computation.** We now describe in more detail how we compute the embeddings shown in Figure 1. We embed and visualize the set $\{X_1, \ldots, X_5, X_{\text{addition}}, X_{\text{intersection}}\}$. We calculate the 768-dimensional embeddings using all-mpnet-base-v2 (Reimers & Gurevych, 2019), which is built into the sentence-transformers package in pytorch. To make these embeddings fit into our framework, we translate them to the nonnegative orthant by applying an additive shift $\mathbf{s} \in \mathbb{R}_{\geq 0}^{768}$ To

do this, we compute the embeddings of the 805 gpt-4o-mini-2024-07-18 outputs from the helpful-base dataset in AlpacaEval (Li et al., 2023). The additive shift $\mathbf{s}$ is taken to be negative of the minimum coordinate along each dimension in this set of 805 embeddings. We translate all 5 outputs and the aggregated outputs by adding $\mathbf{s}$. We compute the variance across the 5 translated outputs vectors along each of the 768 dimensions, and select the top 2 and top 3 dimensions according to variance. We also compute the $\ell_2$-distance between outputs, which is invariant to the additive shift.

