# OpenReview forum: "Power and Limitations of Aggregation in Compound AI Systems"
_ICLR.cc/2026/Conference — ICLR 2026 Conference Withdrawn Submission_

### Official Review · Reviewer_FvLY · 2025-10-21

**Soundness:** 2
**Presentation:** 2
**Contribution:** 1
**Rating:** 0
**Confidence:** 2

**Summary:**

Bad paper.  This paper attempts to present a theoretical framework for understanding why aggregating multiple responses from the same language model can lead to improved performance compared to using a single response.  The authors propose a formalism where each response is modeled as an argmax of a hidden reward function under a set of constraints, representing the model’s internal optimization.  They claim that aggregation leads to better outcomes through one of three mechanisms: **feasibility expansion**, **support expansion**, or **binding set contraction**—that is, the aggregated response lies outside the feasible set of any single response, activates more “coordinates,” or relaxes binding constraints.  The paper also provides several definitions, a “limitation-characterizing condition,” and theorems that restate these mechanisms as necessary or sufficient conditions for aggregation to yield improvement.

Because the paper lacks grounding, insight, and professional presentation, I recommend **rejection**.  It may be salvageable only if the authors (i) re-motivate the setting with realistic assumptions about LMs, (ii) connect their abstract constructs to measurable phenomena, and (iii) include experiments or simulations to demonstrate empirical relevance.  Otherwise, the work remains a purely definitional exercise detached from both theory and practice.

**Strengths:**

- **Ambition and clarity of intent.**
  The paper aims to address an important conceptual question: *why* multi-sample aggregation often improves large language model outputs.  In doing so, it attempts to explain test time compute scaling, a known power empirical LLM phenomena.

- **Formal framing.**
  The authors present clear mathematical definitions and formal statements.  The structure (definitions, theorems, proofs) is coherent.

- **Topic relevance.**
  Aggregation and inference-time scaling are active areas of research in 2025 (e.g., Huang et al., *Is Best-of-N the Best of Them?*, 2025).  Exploring these through theory could be valuable if grounded in actual LM behavior.

**Weaknesses:**

This paper has many problems and reflects bad research taste.

**1. Implausible assumptions and circular abstractions.**
The biggest hindrance to one's comprehension of the paper is that the core modeling assumptions are not defensible:

- Each user prompt is assumed to encode a *hidden reward function* that the model optimizes.
  This is unrealistic both for human communication and for current LM architectures.  Tell me: what is the reward function that I am implicitly implying to you when I'm writing this sentence?
- Each model response is assumed to be the *argmax* of that hidden reward.
  But language models are stochastic samplers, not deterministic solvers of constrained optimization problems.  Furthermore, there's no reason to think LMs are able to always find those argmax.
- Each response is represented as an $M$-dimensional “feature vector” whose coordinates correspond to abstract attributes such as *truthfulness* and *politeness*.  The authors then propose operations such as coordinate-wise min or convex combination over these vectors.
  In other words, if we have two responses, one .9 polite but .1 truthful and the other .1 polite but .9 truthful, then the authors think that there is a way to combine the two responses into something scoring .5 on both truthful and polite dimensions or a response that is .1 polite and .1 truthful.  I am not sure how this combination can be done in practice.  The reason why I am asking ChatGPT a question is obviously that I don't know the true answer to the question.  But in order to precisely take out the truthful portions from the two responses selectively and combine them into one with the right ratio, one probably needs to have some ground truth access to the actual answer in the first place.
- Overall, it's unclear to me how the principal-agent framework (which I personaly do like a lot) can possibly be relevant in this context.

Together these assumptions detach the framework from how LLMs actually operate.

---

**2. No empirical or synthetic validation.**
The paper claims to model mechanisms of improvement in “compound AI systems,” but as one can expect from reading the unrealistic definitions, it presents no data, experiments, or even toy simulations.  This is a pathology with entirely imaginary theoreticians who are not grounded in the real practice of LMs.
No synthetic example demonstrates that the proposed mechanisms reproduce empirically observed best-of-$N$ effects or other aggregation algorithms.
Without grounding, it is impossible to judge whether the theoretical constructs correspond to measurable LLM behavior.

---

**3. Weak or trivial insight.**
Once the readers managed to convince themselves, with the suspension of disbelief, into reading the actual theory, it turns out that the paper contains trivialities.  The main takeaway—“aggregation helps by expanding feasibility/support or relaxing constraints”—really cannot be the most insightful or surprising thing in the world.  The proofs are essentially *definition chasing* and do not reveal deep or counterintuitive explanations about the empirical phenomena.  The framework offers no testable predictions, substantive implications, or connections to practical design of aggregation algorithms.

---

**4. Weird and uncheckable theoretical condition making the proofs easy.**
The “limitation-characterizing condition” (Def. 4.2) is opaque and lacks algorithmic interpretability.  There is no procedure to determine whether real models (e.g., GPT-4o, Gemini-2.5-Pro) satisfy it.  This makes the results unverifiable and undermines the claim that the theorems capture meaningful necessary conditions.  In fact, it appears the condition was reverse-engineered to make the proofs straightforward rather than to capture real constraints.  That also aggreviates the problem that the math here is neither elegant nor hard.

---

**5. No connection to existing aggregation literature.**
The paper ignores prior work that analyzes best-of-$N$ and inference-time alignment empirically and theoretically, such as Huang et al. (2025) *Is Best-of-N the Best of Them?* and the references therein.
These learning theory works model aggregation via *coverage* and *sharpening*, grounded in data.  The mechanisms proposed here—“feasibility expansion,” “support expansion,” “binding contraction”—bear no relationship to known empirical phenomena.  What exactly is the support or the conic constraint when you run Llama3.1 on GSM8K or TruthfulQA?  Indeed, the authors do not appear to be familiar with the relevant literature or the professional norms in learning theory research or AI research.

---

**6. Poor writing and presentation quality.**
The presentation is unprofessional.  The numerous grammatical and English issues distract the reading experience of an otherwise intellectually unchallenging paper.  **Even in the abstract**, there are glaring mistakes suggesting that **the authors didn't read through the manuscript** prior to submission.  The third sentence (“Our analysis uncovers three natural mechanisms—feasibility expansion, support expansion, and binding set contraction—through which aggregation provides benefit to the system designer.”) and fourth sentence (“Our analysis identifies three mechanisms—feasibility expansion, support expansion, and binding set contraction—through which aggregation can expand the set of elicitable outputs.”) are the repetitive rephrases of each other, with a classic double em dash. This happens when someone is over-depenedent on LMs for rephrasing but does not read the resultant content.  There is also a missing space in line 24 before “Altogether.”  If you just read through the abstract once, this wouldn't have happened.  Are you not even reading the abstract before submission now?  Seriously?
- line 58: "We *capture* prompt engineering *limitations* as the *rewards* operating over a coarser M-dimensional feature space."  How can limitations of something be captured as rewards?  The sentences are not making sense.
- line 134 typo on  "hallucinations"
- line 471 subject verb disagreement "Our results offer a theoretical insights"
- line 1009 typo on "elicitability"
The list goes on.  It's no excusable to be so bad with English when the authors have all the language models in the world to help with proofread and edit for expression.  The authors' unprofessionalism hurts readability.

It has not been a productive use of my time to read a paper which the authors didn't care enough to properly prepare.

---

**7. Lack of contribution.**
The paper overstates its significance.  The authors made various baseless claims.
- "...our results uncover key mechanisms that underpin the power and limitations of an aggregation in compound AI systems."
No.  I don't think one knows more about these after reading the paper than before.

- "Our results suggest conditions for aggregation to add no power to a system, regardless of the level of prompt engineering limitations."
The condition is uncheckable, un-verifiable, and has no predictive power.  Can the authors design an experiment where aggregating multiple responses from an LM does NOT help performance, as predicted by the theory?

- "Moreover, our results illustrate how the power of an aggregation depends on the interplay between prompt engineering ability and model capabilities."
There's no clean characterisation of the interplay.  In fact, what even is the quantifiable way to measure prompt engineering ability?  This is once again just some abstract imaginary phrases being thrown around.

- "More broadly, our results take a step towards understanding when aggregation of multiple copies of the same model provides benefits to system designers."
This statement is false.  I will change my mind if the authors can provide three examples of real-world AI system designers who have been benefitted from reading the paper.


In short, there is no experiment, simulation, or practical implication to demonstrate progress toward the stated goals; judging this from a pure mathematics perspective, this would be at the level of math REUs (research experience for undergraduate).  As a pure theory paper, the mathematics are elementary and the results offer little novelty or depth.

**Questions:**

1. How do the authors justify modeling user prompts as reward functions and responses as exact argmaxes?

2. Can the authors provide a concrete mapping between all of their theoretical variables and observable quantities in real LM systems?

3. Why were the aggregation operations restricted to coordinate-wise min and convex combination?
   Have the authors considered realistic aggregation methods such as majority voting, reward-model ranking, self-reward scoring, or log-prob mixture?

4. How could one empirically test whether a model satisfies the “limitation-characterizing condition”?
   Can the authors provide pseudo-code or an algorithmic diagnostic?

5. Would the authors consider adding even a small-scale simulation to demonstrate that the claimed mechanisms can occur in a controlled environment?

6. How do the proposed mechanisms relate to known empirical theories like *sharpening* or *coverage* (Huang et al., 2025)?
   If the relationship is orthogonal, what unique insight does this framework offer?

---

> ### Author Response · Authors · 2025-12-03
> **Response to Reviewer FvLY**
>
> Thanks for the review. We believe that the reviewer may have misunderstood some aspects of the paper, and we clarify key aspects of our paper's content, methodology, and technical contributions below. We think that these clarifications address several of the concerns raised by the reviewer.
>
>
> **Clarification about the relationship with best-of-N sampling.** Our work studies a fundamentally different setting from best-of-N sampling. Specifically, in our framework, the system designer (the principal) designs a set of *different* prompts and aggregates the model responses from these prompts, rather than sampling the model’s responses from a single prompt. See the related work section (L88-L96) for a discussion.
>
> **References on empirical works on aggregation.** We clarify that our original submission already contained a paragraph of references on empirical works on aggregation in the Related Work section (L88-L99); this includes references to multiple inference-time sampling approaches (L89-91). In the revised version,  we’ve added the Huang et al., 2025  best-of-N reference suggested by the reviewer.
>
> **Style of analysis and Model Limitations.** Our work analyzes a stylized mathematical framework to study conceptual questions about AI ecosystems. This builds on a standard methodology followed by a number of papers (e.g., [A], [B], [C], [D]). Like the other papers that carry out this methodology, our model makes a number of simplifying assumptions for mathematical tractability and cleanness. Our original paper (as well as our revised paper) summarizes key model limitations in the Discussion section.
>
> **Writing issues.** We have addressed the typos in the revised version. We clarify that “We capture prompt engineering limitations as the rewards operating over a coarser M-dimensional feature space” was our intended sentence. (In more detail, our framework captures these limitations as the reward specification operating over coarsenings of the output dimensions (as captured by the features) rather than directly on the output dimensions. For more discussion, see our citations example (Section 2.4).)
>
>
> **Conceptual insights.** We added a new section to discuss how our results shed light on when aggregation provides benefit to system designers (Section 5, paragraph titled “Conceptual insights for system designer”).
>
> **Limitation-characterizing condition.**  In the revised version, we clarify how the limitation-characterizing condition connects to strengthened forms of the three natural mechanisms we introduced (feasibility expansion, support expansion, and binding-set contraction). To do this, we added a discussion of this connection (Appendix E.1), and we reformatted the definition of the limitation-characterizing condition (Definition 4.2).
>
> **Technical tools.** While our technical results are simple to state, proving these results required analyzing the multi-dimensional, multi-player game between the principal and agents. For example, we leveraged tools from optimization theory, such as Motzkin’s transposition theorem and strong duality to prove Theorem 4.1. Moreover, our proof of Theorem 4.4 also requires careful and intricate constructions of a feature map.
>
> **Case study.** To provide intuition for how our framework connects to LLM aggregation, we added an example in Section 2.4 of a natural task: using an LLM to generate a list of papers/citations related to a topic. We show how to instantiate our framework in two different ways to capture two different aspects of prompting and aggregation in the context of this task.
>
> **Mechanisms.** Our mechanisms go beyond providing intuition for high-level strategies such as prompt diversity and ensembling; rather, strengthened versions of these mechanisms fully characterize when aggregation provides benefit to a system designer (Section 4). Moreover, the mechanisms we introduce also connect to empirical phenomena of LLMs observed in previous work; we added a discussion of these connections in a paragraph titled ‘Connecting our mechanisms to empirical phenomena’ in the discussion section (Section 5).
>
> [A]  Zhuang, Simon, and Dylan Hadfield-Menell. "Consequences of misaligned AI." Advances in Neural Information Processing Systems 33 (2020): 15763-15773.
>
> [B] Shirali, Ali, Ariel Procaccia, and Rediet Abebe. "The Hidden Cost of Waiting for Accurate Predictions." ICLR 2025.
>
> [C] Ben-Porat, Omer, and Moshe Tennenholtz. "A game-theoretic approach to recommendation systems with strategic content providers." Advances in Neural Information Processing Systems 31 (2018).
>
> [D] Donahue, Kate, and Jon Kleinberg. "Model-sharing games: Analyzing federated learning under voluntary participation." Proceedings of the AAAI Conference on Artificial Intelligence. Vol. 35. No. 6. 2021.

---

### Official Review · Reviewer_qfm4 · 2025-10-30

**Soundness:** 3
**Presentation:** 3
**Contribution:** 3
**Rating:** 6
**Confidence:** 3

**Summary:**

This paper provides a theoretical analysis of aggregation in compound AI systems—cases where multiple instances of the same model are prompted differently and their outputs combined. Using an extended principal–agent framework, the authors formalize when aggregation can enlarge the set of elicitable outputs beyond what a single model can produce. They identify three mechanisms through which aggregation may add power: (1) feasibility expansion; (2) support expansion; (3) binding-set contraction. The paper presents necessary and sufficient conditions (Theorems 3.7, 4.1–4.4) describing when aggregation helps or is provably useless.

**Strengths:**

1. Important question: Understanding the theoretical limits of compound AI systems is timely and conceptually valuable, given the growing use of ensembles, debate, and multi-agent LLM systems.
2. Theoretical clarity: The framework is clean, extending Kleinberg et al. (2019) to include multiple agents and conic feasibility constraints. The three-mechanism taxonomy is intuitive and could guide future empirical work on ensemble design.
3. Useful characterizations: The necessary-and-sufficient conditions in Theorems 4.1–4.4 provide a clear geometric interpretation of when aggregation can and cannot expand elicitability.

**Weaknesses:**

1. Limited novelty of insight: While the formalization is careful, most conclusions reiterate intuitive ideas (“aggregation helps only if it changes feasibility or coverage”). The paper extends prior principal–agent analyses rather than offering genuinely new conceptual understanding of compound AI.
2. Unrealistic assumptions: The theory models models as deterministic optimizers of smooth reward functions, with linear or conic capability constraints and monotone concave rewards. These abstractions are far from how modern LLMs behave (stochastic, discrete, context-dependent). The results therefore may not meaningfully predict behavior of real systems.
3. No empirical validation: There are no simulations or examples showing that these mechanisms actually appear in practice. Even toy numerical demonstrations would help connect the theory to LLM ensembles or debates.
4. Questionable generality. The analysis assumes identical, non-communicating agents. It is unclear whether the results generalize to more realistic compound systems (e.g., heterogeneous models, interactive agents, stochastic sampling).

**Questions:**

1. Under what real conditions do your key assumptions—coarse reward spaces ($N < M$) and linear conic capability constraints ($C \neq 0$)—actually hold for large language models or other AI systems?
2. How robust are the theorems to relaxing these assumptions (e.g., stochastic agents, nonlinear constraints, interactive aggregation)?
3. Empirical research in multi-agents systems with LLMs usually have really complex aggregators. Are the results invariant under different choices of aggregation operators?

---

> ### Author Response · Authors · 2025-12-03
> **Response to Reviewer qfm4**
>
> Thank you for the review.
>
> **Case study.** To provide intuition for how our framework connects to LLM aggregation, we added an example in Section 2.4 of a natural task: using an LLM to generate a list of papers/citations related to a topic. We show how to instantiate our framework in two different ways to capture two different aspects of prompting and aggregation in the context of this task.
>
> **Complex aggregation.** We note that our framework allows for general (e.g., arbitrarily complex or simple ) aggregation operations.
>
> **Mechanisms.** Our mechanisms go beyond providing intuition for high-level strategies such as prompt diversity and ensembling; rather, strengthened versions of these mechanisms fully characterize when aggregation provides benefit to a system designer (Section 4). Moreover, the mechanisms we introduce also connect to empirical phenomena of LLMs observed in previous work; we added a discussion of these connections in a paragraph titled ‘Connecting our mechanisms to empirical phenomena’ in the discussion section (Section 5).
>
> **Conceptual insights.** We added a new section to discuss how our results shed light on when aggregation provides benefit to system designers (Section 5, paragraph titled “Conceptual insights for system designer”).
>
> **Model limitations.** We agree our theoretical framework makes several simplifying assumptions for mathematical tractability. Our original paper (as well as our revised paper) summarizes key model limitations in the Discussion section.
>
> **Extensions to heterogeneous agents.** Many standard approaches, such as prompt ensembling [A] and emerging multi-agent research agents [B], aggregate many copies of the same model. For this reason, our goal is to characterize when aggregation adds power in the case of homogeneous models, and we view heterogeneous models as outside the scope of this paper. Extending our results to heterogeneous agents with different feature mappings and constraint sets would be an interesting direction of future work. With access to heterogeneous models, the power of aggregation is strictly higher since we can additionally leverage the heterogeneity of the models in addition to the differences in responses we can elicit through different prompting to synthesize new outputs through aggregation.
>
> [A] Arora, Simran, et al. "Ask me anything: A simple strategy for prompting language models." ICLR 2023 arXiv preprint arXiv:2210.02441 (2022).
>
> [B] Hadfield, Jeremy, et al. "How we built our multi-agent research system." Anthropic engineering blog, June (2025).

---

### Official Review · Reviewer_tEsp · 2025-11-01

**Soundness:** 3
**Presentation:** 2
**Contribution:** 3
**Rating:** 6
**Confidence:** 3

**Summary:**

The paper introduces a theoretical framework to analyze the aggregation in compound AI systems, where multiple identical AI agents (e.g., LLMs) produce outputs that are then combined. Using a principal–agent model, the paper  studies how aggregation can expand the set of elicitable outputs, i.e., outputs that can be generated by designing suitable reward specifications (e.g., prompts).
The paper identifies three mechanisms through which aggregation may increase expressive power: Feasibility Expansion, Support Expansion, and Binding Set Contraction.

**Strengths:**

The paper presents a mathematical treatment of compound AI aggregation under a principal–agent model.
The theoretical derivations are strong and have clear logical structure.
The proofs look rigorous and complete.
The paper provides  theoretical basis for analyzing emergent properties of LLM ensembles and multi-agent systems. It also provides conceptual tools to assess the limits of prompt ensembling.

**Weaknesses:**

The framework presentation is more theoretical and it is not clear to me how directly the results map to practical LLM systems with complex interactions. The paper could benefit from some experimental validations.
The mathematical exposition looks precise but may not be accessible to readers. The paper could also benefit from more geometric intuition of the three mechanisms.

**Questions:**

1. How do the results change if some assumptions don’t hold, e.g., that rewards are monotone and concave?
2. Does the framework suggest any concrete guidelines for designers on how to ensemble prompts or design rewards in real-world situations?

---

> ### Author Response · Authors · 2025-12-03
> **Response to Reviewer tEsp**
>
> Thank you for the review.
>
> **Case study.** To provide intuition for how our framework connects to LLM aggregation, we added an example in Section 2.4 of a natural task: using an LLM to generate a list of papers/citations related to a topic. We show how to instantiate our framework in two different ways to capture two different aspects of prompting and aggregation in the context of this task.
>
> **Conceptual insight.** We discuss how our results shed light on when aggregation provides benefit to system designers (Section 5, paragraph titled “Conceptual insights for system designers”). Specifically, aggregation can be useful even as model capabilities improve, due to prompt engineering limitations, and how some aggregation operations offer no power regardless of the level of prompt engineering limitations.
>
> **Mechanisms.** The mechanisms we introduce also connect to empirical phenomena of LLMs observed in previous work; we added a discussion of these connections in a paragraph titled ‘Connecting our mechanisms to empirical phenomena’ in the discussion section (Section 5).

---

### Official Review · Reviewer_H3s3 · 2025-11-01

**Soundness:** 3
**Presentation:** 3
**Contribution:** 2
**Rating:** 4
**Confidence:** 3

**Summary:**

This paper studies compound AI systems through a principal-agent framework, analyzing when aggregating outputs from multiple copies of the same model can elicit a broader set of outputs than querying a single model. The authors extend Kleinberg et al.'s (2019) framework by introducing conic constraints to model capability limitations and characterize three mechanisms through which aggregation provides benefits: feasibility expansion, support expansion, and binding set contraction. The main theoretical contributions include necessary and sufficient conditions for when aggregation expands the set of elicitable outputs.

**Strengths:**

- The paper addresses a current and growing trend: leveraging multiple (often identical) copies of LLMs in a system to increase power, robustness, or reliability, which is relevant in both research and industry.
- Provides clear formalizations and mathematical analysis, precisely characterizing when aggregation of model outputs can or cannot expand the set of desired system-level outputs.
- Identifies three distinct mechanisms (feasibility expansion, support expansion, binding set contraction) and connects these directly to improvements from aggregation, offering useful conceptual tools.
- The paper is clear about which settings aggregation is strictly beneficial, and is precise about which mechanisms are merely necessary versus sufficient.

**Weaknesses:**

- The principal-agent model, as used in Kleinberg et al., are used for classification settings (like job applicants and employers) assumes agents are strategic (with natural game-theoretic incentives, e.g., job applicants maximizing personal outcome). In contrast, LLM “agents” in this paper are neither strategic nor adversarial—they are collaborative and operate according to deterministic/reward-guided rules specified by a system designer. This difference is significant: the economic intuition motivating principal-agent theory (such as strategic effort and misaligned incentives) is not clearly transferred to the compound LLM setting. It is not clear whether all insights from principal-agent theory apply, or if a simpler optimization framework would suffice.

- Homogeneous Setting Only (Limited Generality):
The study focuses almost exclusively on aggregating identical models (“copies of the same model prompted differently”). Many real-world compound systems employ heterogeneity, i.e., varying model sizes, architectures, or capabilities. The theoretical results, as currently framed, may not inform those important cases. The work would benefit from discussion or analysis of how aggregation behaves with heterogeneous agents, particularly as “limitations” may be non-uniform.

- The paper treats reward as a general, abstract “specification” that influences outputs. In practical LLM systems, “reward” is often implicit in prompts, user feedback, or RL-from-human feedback, but not explicit or directly adjustable as in standard economic settings. The discussion would benefit from concrete examples or instantiations of how such “reward” appears in real LLM deployments. Also, bigger/better models will face fewer limitations even with the same prompt/reward specification—the model class is not fixed in practice, undermining a key modeling assumption.

- While the paper is mathematically elegant, it is not clear how the theoretical results translate into actionable guidance for practitioners designing compound LLM systems. For example, are there concrete aggregation rules or mechanisms suggested by the results that outperform naive ensembling or majority voting? Does the result motivate any new kind of prompt engineering, or guidance on when aggregation won’t help? The paper should provide illustrative empirical examples or real-world case studies since otherwise the practical impact is highly limited.

- The main mathematical model assumes that limitations (in reward/prompt and in model output) are all linear or conic for tractability. However, LLM behaviors (like hallucinations and edge-case logical errors) are highly non-linear and context-dependent. Modeling these complex sources of error and capacity via simple linear constraints may not capture the true limitations of deployed LLMs, and might mislead system designers about when aggregation will actually help.

- Not clear this captures real prompt engineering challenges like ambiguity in natural language prompts, inability to specify complex constraints, Prompt brittleness.

- While the paper provides a taxonomy of aggregation mechanisms, it mainly offers justification for why existing conceptual ideas (like “prompt diversity” or “ensemble voting”) sometimes work. It does not provide novel principles or mechanisms for designing better compound systems; rather, it mainly delineates when basic strategies succeed or fail.


Typos:
- repeated line in the abstract
- missing space after comma multiple times in the paper (line 482
- hallunications -> hallucinations

**Questions:**

In addition to the ones above in the weaknesses section:

- Can you provide a concrete example mapping real LLM outputs (e.g., text completions) to your R^M output space? How would one extract the constraint matrix C and feature weights \alpha from a real LLM?

- Can results extend to heterogeneous models? What difference does it make when the agents are strategic vs collaborative (like argmax formulation for aggregation)?

---

> ### Author Response · Authors · 2025-12-03
> **Response to Reviewer H3s3**
>
> Thank you for the review.
>
> **Case study.** To provide intuition for how our framework connects to LLM aggregation, we added an example in Section 2.4 of a natural task: using an LLM to generate a list of papers/citations related to a topic. We show how to instantiate our framework in two different ways to capture two different aspects of prompting and aggregation in the context of this task.
>
> **Conceptual insights.** We added a new section to discuss how our results shed light on when aggregation provides benefit to system designers (Section 5, paragraph titled “Conceptual insights for system designer”).
>
> **Mechanisms.** Our mechanisms go beyond providing intuition for high-level strategies such as prompt diversity and ensembling; rather, strengthened versions of these mechanisms fully characterize when aggregation provides benefit to a system designer (Section 4). Moreover, the mechanisms we introduce also connect to empirical phenomena of LLMs observed in previous work; we added a discussion of these connections in a paragraph titled ‘Connecting our mechanisms to empirical phenomena’ in the discussion section (Section 5).
>
> **How our model captures realistic prompting limitations.** Our framework captures reward specification limitations arising from (1) the system designer struggling to precisely express what they truly want in the prompt, and (2) the model struggling to correctly interpret the system designer's prompt. We capture both of these forms of limitations as the reward specification operating over coarsenings of the output dimensions (as captured by the features) rather than directly on the output dimensions. For more discussion, see our citations example (Section 2.4).
>
> **Applicability of principal-agent framework.** While the principal-agent model in Kleinberg et al., 2019 is motivated by human strategic effort, principal-agent frameworks have been used to capture much more general forms of reward misalignment. For example, Zhuang and Hadfield-Menell [A] also leverage a principal-agent framework to capture reward misalignment between a system designer’s reward and an AI agent’s reward, focusing on underspecification. Borrowing inspiration from [A], we adopt a principal-agent framework -agent framework to capture more general forms of reward misalignment.
>
> **Bigger models.** We agree that bigger/better models face fewer limitations. This is captured in our framework via the constraints C; we envision that “better” models will have fewer constraints. User-side prompt limitations could still persist and aggregation would help mitigate this.
>
> **Extensions to heterogeneous agents.** Many standard approaches, such as prompt ensembling [B] and emerging multi-agent research agents [C], aggregate many copies of the same model. For this reason, our goal is to characterize when aggregation adds power in the case of homogeneous models, and we view heterogeneous models as outside the scope of this paper. Extending our results to heterogeneous agents with different feature mappings and constraint sets would be an interesting direction of future work. With access to heterogeneous models, the power of aggregation is strictly higher since we can additionally leverage the heterogeneity of the models in addition to the differences in responses we can elicit through different prompting to synthesize new outputs through aggregation.
>
> [A] Zhuang, Simon, and Dylan Hadfield-Menell. "Consequences of misaligned AI." Advances in Neural Information Processing Systems 33 (2020): 15763-15773.
>
> [B] Arora, Simran, et al. "Ask me anything: A simple strategy for prompting language models." ICLR 2023.
>
> [C] Hadfield, Jeremy, et al. "How we built our multi-agent research system." Anthropic engineering blog, June (2025).

---

### Author Response · Authors · 2025-12-03
**General response**

Thank you to all of the reviewers for their reviews. We have revised our paper to incorporate reviewer feedback. We’ve marked the main changes in blue, and we summarize these changes below:

**Illustrative example to demonstrate our model (Section 2.4).** To provide intuition for how our framework connects to LLM aggregation, we added an example of a natural task: using an LLM to generate a list of papers/citations related to a topic.
We show how to instantiate our framework in two ways to capture two different aspects of prompting and aggregation in the context of this task. For one of these instantiations, we provide an empirical visualization of high-dimensional embeddings which shows how different aggregation rules based on GPT-4o can produce semantically different outputs.
We also describe how our framework captures reward specification limitations arising from (1) the system designer struggling to precisely express what they truly want in the prompt, and (2) the model struggling to correctly interpret the system designer's prompt. We capture both of these forms of limitations as the reward specification operating over coarsenings of the output dimensions (as captured by the features) rather than directly on the output dimensions.

**Connection between limitation-characterizing condition and mechanisms (Appendix E.1).** We clarify how the limitation-characterizing condition connects to strengthened forms of the three natural mechanisms we introduced (feasibility expansion, support expansion, and binding-set contraction). To do this, we added a discussion of this connection (Appendix E.1), and we reformatted the definition of the limitation-characterizing condition (Definition 4.2).

**Conceptual takeaways for designers.** We discuss how our results shed light on when aggregation provides benefit to system designers (Section 5, paragraph titled “Conceptual insights for system designers”). Specifically, aggregation can be useful even as model capabilities improve, due to prompt engineering limitations, and how some aggregation operations offer no power regardless of the level of prompt engineering limitations.

**Connecting our mechanisms to empirical phenomena.**  The mechanisms we introduce connect to empirical phenomena of LLMs observed in previous work. We added a discussion of these connections in a paragraph titled ‘Connecting our mechanisms to empirical phenomena’ in the discussion section (Section 5). We believe that these connections can point toward empirical settings where aggregation adds power, and we view empirical validation of our results as an interesting direction for future research.

We would also like to reiterate that our theoretical framework makes several simplifying assumptions for mathematical tractability; our original paper (as well as our revised paper) summarizes key model limitations in the Discussion section (Section 5, paragraph titled “Model limitations and extensions”).

---

### Note · Authors · 2025-12-31

**Comment:**

Thank you for reviewing our paper! We appreciated your feedback and will use it to improve the paper for future submissions.

**Withdrawal Confirmation:**

I have read and agree with the venue's withdrawal policy on behalf of myself and my co-authors.